



# Natural sea-salt emissions moderate the climate forcing of anthropogenic nitrate

Ying Chen[1,2,3,8,*], Yafang Cheng[2,*], Nan Ma[4,2,3,1], Chao Wei[3], Liang Ran[5], Ralf Wolke[1], Johannes Größ[1], Qiaoqiao Wang[4], Andrea Pozzer[6], Hugo A. C. Denier van der Gon[7], Gerald Spindler[1], Jos Lelieveld[6,9], Ina Tegen[1], Hang Su[3], Alfred Wiedensohler[1]

[1]Leibniz-Institute for Tropospheric Research, Leipzig, Germany
[2]Minerva Research Group, Max Planck Institute for Chemistry, Mainz, Germany
[3]Multiphase Chemistry Department, Max Planck Institute for Chemistry, Mainz, Germany
[4]Center for Pollution and Climate Change Research (APCC), Institute for Environmental and Climate Research, Jinan University, Guangzhou, China
[5]Key Laboratory of Middle Atmosphere and Global Environment Observation, Institute of Atmospheric Physics, Chinese Academy of Sciences, Beijing, China
[6]Atmospheric Chemistry Department, Max Planck Institute for Chemistry, Mainz, Germany
[7]TNO, Dept. of Climate, Air and Sustainability, Princetonlaan 6, Utrecht, the Netherlands
[8]Lancaster Environment Centre, Lancaster University, Lancaster, UK
[9]The Cyprus Institute, Nicosia, Cyprus

*Correspondence to:* yafang.cheng@mpic.de (Y.F.C.); chen@tropos.de (Y.C.)

**Abstract.** Natural sea-salt aerosols, when interacting with anthropogenic emissions, can enhance the formation of particulate nitrate. This enhancement has been suggested to increase the direct radiative forcing of nitrate, called 'mass-enhancement effect'. Through a size-resolved dynamic mass transfer modelling approach, we show that interactions with sea-salt shift the nitrate from sub- to super-micron sized particles ('re-distribution effect'), and hence lower its efficiency for light extinction and reduce its lifetime. The re-distribution effect overwhelms the mass-enhancement effect and significantly moderates nitrate cooling; e.g., the nitrate associated aerosol optical depth can be reduced by 10-20% over European polluted regions during a typical sea-salt event, in contrast to an increase by ~10% when only accounting for the mass-enhancement effect. Global model simulations indicate significant re-distribution over coastal and offshore regions world-wide. Our study suggests a strong buffering by natural sea-salt aerosols that reduces the climate forcing of anthropogenic nitrate, which had been expected to dominate the aerosol cooling by the end of the century. Comprehensive considerations of this re-distribution effect foster better understandings of climate change and nitrogen deposition.



## 1. Introduction

Particulate nitrate ($NO_3^-$) is one of the most important anthropogenic aerosol components that exert a climate cooling effect (IPCC, 2013;Haywood and Schulz, 2007). On a global scale, its average direct radiative forcing (DRF) has been estimated to span over a relatively wide range from -0.08 to -0.19 W m$^{-2}$ (Liao et al., 2004;Liao and Seinfeld, 2005;IPCC, 2013;Xu and Penner, 2012;Haywood and Schulz, 2007;Myhre et al., 2013;Forster et al., 2007;Adams et al., 2001;Bauer et al., 2007;Jacobson, 2001;Streets et

al., 2013;van Dorland et al., 1997). It is projected to reach up to about -0.4 to -1.3 W m$^{-2}$ and dominate the aerosol cooling by the end of the century (Adams et al., 2001;Bellouin et al., 2011;Hauglustaine et al., 2014). Although the gaseous precursors of nitrate, e.g., NOx (= NO + NO$_2$) and HNO$_3$, are mainly of anthropogenic origin, globally about 35-50% of the nitrate mass is associated with natural sea-salt aerosol (Xu and Penner, 2012;Myhre et al., 2006). This is because sea-salt aerosol can be transported over industrialized regions,

interacts with anthropogenic precursors of nitrate and enhances the total nitrate column loading in the atmosphere through heterogeneous uptake of HNO$_3$ and its precursors (Liao et al., 2004;Liao and Seinfeld, 2005;Seinfeld and Pandis, 2006;Xu and Penner, 2012;Ravishankara, 1997;Lowe et al., 2015). Such sea-salt-induced nitrate mass increase is believed to strengthen the DRF and climate cooling of nitrate (Liao and Seinfeld, 2005), called the 'mass-enhancement effect'.

However, not only the mass concentration but also the particle mass size distribution (PMSD) are essential for evaluating the direct radiative forcing of nitrate (DRF$_{nitrate}$) (IPCC, 2013;Murphy et al., 1998;Kok et al., 2017). Besides leading to the increase of total nitrate mass, interactions with sea-salt aerosol also have a 're-distribution effect' on nitrate PMSD which shifts nitrate from sub- to super-micron sizes (e.g., Chen et al., 2016a, see also Fig. 1). Because sea-salt aerosol is mainly present as super-micron (coarse)

particles (Murphy et al., 1998;O'Dowd et al., 1997;Ravishankara, 1997), chemical equilibrium favors the formation of thermodynamically stable sodium nitrate in the coarse mode, which inhibits the formation of semi-volatile ammonium nitrate in the sub-micron size (fine mode) through competitive consumption of gaseous precursors and change of gas-particle equilibrium (Chen et al., 2016a;Zaveri et al., 2008;Myhre et al., 2006). Compared to the fine particles, coarse particles have a significantly lower extinction efficiency in

the visible part of the spectrum (IPCC, 2013;Murphy et al., 1998), the sea-salt-induced 're-distribution effect' thus tends to weaken DRF$_{nitrate}$, which counteracts the 'mass-enhancement effect'. The competition between these two effects will ultimately determine the net impact of sea-salt aerosol on anthropogenic DRF$_{nitrate}$. To





the best of our knowledge, this sea-salt-induced 'γe-distribution effect' is neglected in most global models, due to the computationally expensive calculations of fully dynamic mass transfer between size-resolved

particulate nitrate (Adams et al., 2001;Myhre et al., 2006); for more details see Supplementary Information Section S1.

To explore the competition between the sea-salt-induced 'mass-enhancement effect' and 're-distribution effect' and the impact of the 're-distribution' process on the nitrate cooling of climate, we conducted a series of sensitivity studies with and without sea-salt aerosol emission for a typical sea-salt event over Europe and

North America, using a regional atmospheric chemistry model (WRF-Chem) with a fully dynamic mass transfer approach (Zaveri et al., 2008); see 'Data & Methods' for details. A one-year simulation with the EMAC (ECHAM5/MESSy Atmospheric Chemistry) model is used to demonstrate the importance of the 're-distribution effect' on a global scale (Jöckel et al., 2010).

## 2. Data & Methods

### 2.1 Observations

HOPE-Melpitz campaign (HD(CP)[2] Observational Prototype Experiment, Macke et al., 2017) was carried out during 10-20 September 2013 at Melpitz (12.93°E, 51.53°N, 86 m a.s.l.). Melpitz represents the regional background of central Europe (Spindler et al., 2012), with flat surrounding topography over an area

of hundreds of square kilometers, ranging 100-250 m a.s.l.

Size-segregated measurements of particles composition were carried out on 13 September and 18 September, which represent the continental period and marine period, respectively. A five-stage Berner impactor (Hauke, Austria, 0.05-0.14 $\mu$m, 0.14-0.42 $\mu$m, 0.42-1.2 $\mu$m, 1.2-3.5 $\mu$m, and 3.5-10 $\mu$m; Berner and Luerzer, 1980) was operated to segregate particles onto ring-like pre-baked (350 °C) aluminium foils

with a sampling period of 24 h (00:00-24:00, local time) for detailed chemical analyses (Spindler et al., 2012). The isokinetic inlet for particles with an aerosol dynamic diameter smaller than 10 $\mu$m was installed 6 meters above the ground. To compare with modelling results, we use the sum of the particle mass at stage 1-3 (PM$_{1.2}$, aerosol dynamic diameter smaller than 1.2 $\mu$m) to represent fine mode particles and the sum of the mass at stages 4-5 (PM$_{1.2-10}$, aerosol dynamic diameter smaller between 1.2 and 10 $\mu$m) to represent

coarse mode particles. The gravimetric mass of the pre-heated aluminium foils was weighted respectively



before and after the sampling process, by a microbalance (UMT-2, Mettler-Toledo, Switzerland). Before each weighting, the aluminium foils were equilibrated for at least 72 hours in a strictly controlled environment with a temperature of 20±1 ℃ and a relative humidity of 50±5%. After an aqueous extraction of foil aliquots, the main water-soluble cations ($Na^+$, $NH_4^+$, $K^+$, $Mg^{2+}$ and $Ca^{2+}$) were quantified by standard

ion chromatography (Neusüß et al., 2000). Likewise, capillary electrophoresis (Neusüß et al., 2000) was carried out to quantify the anions ($NO_3^-$, $SO_4^{2-}$ and $Cl^-$). A carbon analyzer (Behr Labor-Technik, Germany) was used to separate and measure the sampled organic and elemental carbon with a two-step thermographic method (modified VDI method 2465 part-2, Spindler et al., 2012). Organic carbon was vaporized at 650 ℃ for 8 minutes under $N_2$ and catalytically converted to $CO_2$; the remaining elemental carbon was combusted

for another 8 minutes with $O_2$ at 650 ℃. Generated $CO_2$ was then quantitatively determined using a non-dispersive infrared detector.

## 2.2 Model description

     We performed regional model simulations with the 'online coupled' air quality model Weather Research

and Forecasting/Chemistry model (WRF-Chem V3.5.1, Grell et al., 2005). WRF-Chem enables more detailed investigation of aerosol-radiation interaction over specific regions at higher horizontal resolution compared with global models, and has been broadly used for investigating aerosol radiative forcing in previous studies (e.g., Archer-Nicholls et al., 2019;Fast et al., 2006;Saide et al., 2012;Gao et al., 2018;Yao et al., 2017;Huang et al., 2015). To investigate the impact of 're-distribution effect' on PMSD and climate

effect of nitrate, the fully dynamic aerosol module MOSAIC (Zaveri et al., 2008) was utilized with eight discrete size bins (39-78 nm, 78-156 nm, 156-312 nm, 312-625 nm, 625-1250 nm, 1.25-2.5 $\mu$m, 2.5-5 $\mu$m, 5-10 $\mu$m; see also Fig. S1), with the online coupled CBMZ (Carbon-Bond Mechanism version Z) gas chemistry scheme (Zaveri and Peters, 1999). The sea-salt emissions computed with the modified Gong scheme (Gong, 2003) were reduced to 10% in the 'Case_SeasaltOn', because a previous study (Chen et al.,

2016a) has shown that the original Gong scheme overestimates the sea-salt mass concentrations by a factor of ~10 over the coastal regions of Europe. The sea-salt emission was turned off in the 'Case_SeasaltOff' simulation. We calculate the DRF of anthropogenic nitrate at the top of the atmosphere for both with and without sea-salt presence respectively, based on the difference in the net incoming radiative flux with and without the anthropogenically emitted gas phase precursor NOx (IPCC, 2013;Xu and Penner, 2012). Only



the model results during daytime (07:00-16:00, local time) and under clear-sky condition (cloud optical

depth equals to zero) were used for the analyses of DRF in this study.

WRF-Chem calculated aerosol optical depth (AOD) and direct radiative effect of total aerosols based

on the internal mixture assumption and taking the hygroscopicity into account. In order to calculate the light

extinction coefficient and optical depth for individual aerosol species, we performed the following off-line

calculation. The AOD of each species is calculated by integrating light extinction coefficient ($\sigma_{ex}$) over all

vertical layers. The $\sigma_{ex}$ of sea-salt (NaCl) and particulate nitrate were calculated with Mie theory, based on

their PMSD. Different from the WRF-Chem calculation of total aerosol AOD, external mixture was assumed

for nitrate and sea-salt particles when calculating their respective contributions on $\sigma_{ex}$ or AOD. Hygroscopic

growth was also considered, following the κ-Köhler theory (Köhler, 1936;Petters and Kreidenweis, 2007).


**2.3 European simulation**

The European simulations focus on the HOPE-Melpitz campaign period of 10-20 September 2013,

during which a sea-salt event that influenced most part of Europe was captured. The simulations are defined

by two nested domains with horizontal resolutions of 54 km and 18 km respectively, and 39 vertical layers

with model top at 50 hPa. The coarse domain (D01, $30^o$ N – $71.5^o$ N, $30^o$ W – $46^o$ E) covers the North Sea,

the European continent and part of North Africa. The inner domain (D02, $38^o$ N – $60^o$ N, $8^o$ W – $28^o$ E, see

Fig. 2) covers most of the North Sea and the European continent. The European anthropogenic emission

inventories are provided by TNO (www.tno.nl) from the AQMEII project (Air Quality Model Evaluation

International Initiative) for $PM_{2.5}$, $PM_{2.5-10}$, NOx, $SO_2$, CO, $NH_3$, and non-methane volatile organic

compounds (Pouliot et al., 2012;Chen et al., 2018b), and from the EUCAARI project (European Integrated

project on Aerosol, Cloud, Climate, and Air Quality Interactions) for particulate organic carbon and

elemental carbon (Kulmala et al., 2011). The inventories are with a spatial resolution of $1/8^o \times 1/16^o$

longitude-latitude. We excluded the point source emissions of elemental carbon in EUCAARI inventory over

Germany, since a previous study (Chen et al., 2016b) reported large uncertainty of them. More details about

emissions, meteorological and chemical initial/boundary conditions can be found in the Supplementary

Information Section S2. Detailed information about the model configuration is given in Table 1.

In the 'Case_SeasaltOn' (with sea-salt emission) of European simulation, modelled [Na$^+$] showed good

agreement with measurements over coastal region from European Monitoring and Evaluation Programme



(EMEP, http://www.emep.int), with a factor (and correlation coefficient) of 0.85 (0.67), 1.16 (0.80) and 0.83

(0.87) respectively for Bilthoven, Kollumerwaard and Vredepeel (Fig. S2). Compared with ground-based

measurements at Melpitz and radio-sounding measurements across Europe, the meteorological conditions

were well captured by the model (Chen et al., 2016a). Vertical structures of potential temperature and wind

speed were realistically reproduced, with correlation coefficients between simulation and measurement

results of ~0.9 over coastal, German low lands (Melpitz) and northern Poland regions (Chen et al., 2016a).

In line with previous studies (Xu and Penner, 2012;Li et al., 2013), the modelled AOD agreed reasonably

well with the AERONET observations (AErosol RObotic NETwork, http://aeronet.gsfc.nasa.gov). The

spatial distribution of AOD can be generally captured by the model (R=0.64), although model may

overestimate AOD by a geometric mean bias of 70% (see Supplementary Information Section S3 for details).

**2.4 North America and global simulations**

To investigate the significance of the 're-distribution effect' in a boarder spatial scale, we also

conducted WRF-Chem simulation over North America, where high concentration of nitrate was usually

observed. We focus on the period of 10-17 January 2015, when strong continental outflow interacted with

marine air masses over the Gulf of Mexico. The North American domain covers US, the Gulf of Mexico and

part of Pacific and Atlantic oceans, with a horizontal resolution of 36 km. In addition, a one-year simulation

with global model (EMAC) was carried out for analysis of the potential impact of 're-distribution effect' on

a global scale, although the fully dynamic mass transfer between particle sizes is not considered in EMAC

(four size modes rather than eight size bins as applied in the WRF-Chem model). More details of EMAC

model and its aerosol module are described in our previous work (Pozzer et al., 2012;Klingmüller et al.,

2014;Pringle et al., 2010). The global simulation was run at T106L31 resolution, corresponding to a

quadratic Gaussian grid of approximately 1.1 by 1.1 degrees (in latitude and longitude) and with 31 levels

in the troposphere. The global emission inventory EDGAR (V4.3, 2010, http://edgar.jrc.ec.europa.eu) was

used in the North American and global simulations.

**3. Results and Discussion**

**3.1 Sea-salt-induced 're-distribution effect'**

Marine air masses frequently (~90%) approach Central Europe, and the interaction between



anthropogenic pollutants and sea-salt aerosol commonly happens in the atmosphere (Birmili et al., 2001). In this study, we performed a series of numerical sensitivity experiments during a typical sea-salt transport

event from 10 to 20 September 2013 during the HOPE-Melpitz campaign. During the campaign, Central Europe was dominated by continental air masses before 15 September. Subsequently marine air masses started travelling over land on 17 September, and sea-salt aerosol originated from the North Sea and the Baltic Sea was transported to northern Poland and dominated Central Europe on 19 September (Fig. 2). Sea-salt is emitted into the marine planetary boundary layer (PBL) as coarse particles, usually with a short

lifetime and a limited transport range. Previous studies showed that the special PBL thermodynamic structure over coastal regions (Ding et al., 2004) can bring sea-salt from the marine PBL to the continental free troposphere, therefore prolonging its lifetime and favoring long-range transport, see the figure 11 of Chen et al. (2016a). Afterwards, the sea-salt aloft could be mixed down to surface layer by the fully developed PBL (Chen et al., 2009;Chen et al., 2016a) and interact with anthropogenic nitrate. Therefore, this transport

mechanism broadens the sea-salt-induced 're-distribution effect' on nitrate to a larger spatial scale.

As shown in Fig. 3a and 3b, the 'Case_SeasaltOn' simulation successfully reproduces the 're-distribution effect' of nitrate when the dominating air mass changed from a continental to marine type at the Central Europe background site Melpitz. Measurements and model results both show an ~10 times increase of sodium concentration ($[Na^+]$, indicator of sea-salt aerosol) in the coarse mode when marine air masses

approached (Fig. S3). While the nitrate mass fraction in the coarse mode was enhanced by a factor of ~5.5, its concentration ($[NO_3^-]$) in the fine mode was lowered by ~20% (Fig. 3b). A clear re-distribution of particulate nitrate from fine to coarse mode is found when introducing sea-salt in the 'Case_SeasaltOn', with negligible changes in other particulate species except sea-salt (Fig. S1). Conversely, without introducing sea-salt aerosol, the 'Case_SeasaltOff' did not capture the 're-distribution effect', and the nitrate mass in the fine

mode (~97%) dominated in both continental and marine air masses (Fig. 3c).

**3.2 Moderation of nitrate cooling by the 're-distribution effect'**

Figure 4 shows the strong impact of the 're-distribution effect' on nitrate cooling when marine air mass transported further inland and predominated over Europe on 19 September. Although sea-salt aerosol leads

to an overall enhancement of nitrate column loading (~1 mg m$^{-2}$, Fig. S4) compared to the 'Case_SeasaltOff', the nitrate associated aerosol optical depth (AOD$_{nitrate}$) decreases significantly over the relatively polluted



continental regions (Fig. 4a), leading to a strongly weakened cooling effect of nitrate in those regions (positive change in $DRF_{nitrate}$ in Fig. 4b). We find that the regions of reduced $AOD_{nitrate}$ co-locate with the regions of reduced fine mode $[NO_3^-]$ (bluish colored areas in Fig. 4a and 4c). Over a large area of the

European continent, the 're-distribution effect' shifts the nitrate PMSD from the fine to the coarse mode (Fig. 4c, 4d), resulting in much less efficient light scattering of nitrate aerosol with a reduced cooling effect. The box with a solid black frame in Fig. 4 marks the northern Poland region, where sea-salt aerosol strongly interacted with anthropogenic precursors of nitrate during the studied sea-salt event. In this region, 're-distribution effect' leads to a decrease of column nitrate loading in the fine mode by ~2.9 mg m$^{-2}$ (Fig. 4c)

accompanied by an increase of ~3.7 mg m$^{-2}$ in the coarse mode due to the 'mass-enhancement effect' (Fig. 4d). Consequently, the anthropogenic $AOD_{nitrate}$ is significantly reduced by up to ~30% with an average reduction of ~22% (~0.05 in absolute AOD value, Fig. 4a), despite of a ~0.8 mg m$^{-2}$ net increase in total nitrate loading. This results in a 5-70% reduction of nitrate cooling, on average by ~26% (Fig. 4b). Our results demonstrate that the sea-salt-induced 're-distribution effect' overwhelms the 'mass-enhancement

effect' over polluted regions, thus moderating the anthropogenic nitrate cooling.

The resulting decrease of nitrate cooling is non-linear with respect to the sea-salt aerosol loading due to the competition between sea-salt-induced 'mass-enhancement effect' and 're-distribution effect'. To investigate the net impact of this competition, we performed a series of sensitivity simulations with different sea-salt aerosol emission scaling factors (0, 0.5, 1, 2, 3, 4, 5, 6, 8, and 10, the ones with scaling factors of 0

and 1 being our 'Case_SeasaltOff' and 'Case_SeasaltOn', respectively). As a surrogate of aerosol DRF, the AOD of nitrate and sea-salt aerosol was calculated offline based on the simulated PMSD (see 'Data & Methods'). Over the northern Poland region (box with solid frame in Fig. 4), total $[NO_3^-]$ (green dashed line in Fig. 5a) increased continuously as a function of $[Na^+]$, which is in line with previous estimates and clearly shows the 'mass-enhancement effect' (Liao and Seinfeld, 2005). However, the 're-distribution effect'

overwhelms the 'mass-enhancement effect' in this region, and the $AOD_{nitrate}$ drops significantly by ~29% (red solid line in Fig. 5a) when the scaling factor of sea-salt aerosol is 1 ('Case_SeasaltOn', being slightly higher than the average ~22% from the on-line calculation, due to the difference in aerosol mixing state, i.e., internal mixing of aerosol compositions in the on-line calculation; and external mixing in the offline calculation). The reduction of $AOD_{nitrate}$ reaches a maximum of ~50% (~0.09 in absolute AOD value) when

$[Na^+]$ is ~2.5 $\mu$g m$^{-3}$ or higher, which level of sea-salt aerosol has been often observed in Central Europe





(Fig. 5c, Gustafsson and Franzén, 2000;Neumann et al., 2016;Gantt et al., 2015). A similar net reduction of AOD$_{nitrate}$ is also found for Central Europe (marked in Fig. 2), where the overall moderation is ~13% compared to the 'Case_SeasaltOff' (Fig. S5a). To further demonstrate the influence of the 're-distribution effect' on nitrate cooling, we calculate the ΔAOD$_{nitrate}$ by re-allocating nitrate mass into different size bins

according to the normalized nitrate PMSD simulated in the 'Case_SeasaltOff', i.e., by neglecting the 're-distribution effect' (ΔAOD$^*_{nitrate}$, pink dashed line in Fig. 5a). A distinct opposite trend, with increasing AOD$^*_{nitrate}$ with [NO$_3^-$] (green dashed line), would result by neglecting the 're-distribution effect'. For example, instead of a decrease by 29%, the AOD$^*_{nitrate}$ over northern Poland increases by ~8% from the 'Case_SeasaltOff' to the 'Case_SeasaltOn'.

It is also noteworthy that the increase rate of [NO$_3^-$] decreases as [Na$^+$] further increases and there is even a slight decrease of [NO$_3^-$] when [Na$^+$] exceeds ~5 $\mu$g m$^{-3}$ (Fig. 5a). This is mostly due to the consumption of nitrate precursor and stronger deposition of particulate nitrate by 're-distribution' toward larger particles sizes and thus shorter lifetimes. This result is consistent with previous studies showing that lifetime of nitrate radical is significantly reduced in the presence of sea-salt aerosols (Rudich et al.,

1998;Ravishankara, 1997). The lifetime of nitrate particles can be shortened from about a week to one day by shifting from the fine to the coarse mode (Croft et al., 2014;Chen et al., 2016b) and hence further moderate nitrate cooling effect. Moreover, coarse mode nitrate associated with highly hygroscopic sea-salt enhances its water uptake (Chen et al., 2018a) and cloud condensation nuclei activation (Xu and Penner, 2012;Wang and Chen, 2019), therefore fasten its deposition and scavenge rate. Itahashi et al. (2016) reported that

oxygenated nitrogen deposition can be enhanced by 1.6-2.2 times over ocean regions in East Asia by including sea-salt associated nitrate. Liao and Seinfeld (2005) implied a similar enhancement for sea-salt/dust associated sulfate deposition, which result in a decrease in sulfate concentrations in the downwind regions. Our study implies the enhancement of nitrate deposition. As one can see in Fig. 5a, nitrate concentration (green dashed line) firstly increases with increasing sea-salt and then slightly declines when

the amount sea-salt (represented by sodium) exceeds a certain level. An inflection point is observed when sodium concentration approached 5.5 $\mu$g m$^{-3}$ (sea-salt emission factor = 8), the nitrate starts to decrease as sodium further increases (sea-salt emission factor = 10). The enhancement of nitrate deposition should be the reason of this phenomenon. To demonstrate this, we conducted a sensitivity simulation with aerosol dry deposition turned off. We found that nitrate concentration kept increasing from 10.91 to 11.02 $\mu$g m$^{-3}$ when





sea-salt emission factors increased from 8 to 10, instead of showing a decreasing trend as the simulations

with aerosol dry deposition. This indicates an enhanced deposition of nitrate by interaction with sea-salt

which shortens nitrate lifetime and can further reduce nitrate cooling.

**3.3 Competition between 're-distribution effect' and 'mass-enhancement effect'.**

As a result of competition between the sea-salt-induced 're-distribution effect' and 'mass-enhancement

effect', clear spatial inhomogeneity can be found in Fig. 4. The 're-distribution effect' decreases $AOD_{nitrate}$

over the polluted continental regions; while the 'mass-enhancement effect' increases $AOD_{nitrate}$ mostly over

marine areas (Fig. 4a), although the absolute enhancement is rather small due to the low $[NO_3^-]$ in this

environment. For example, over the Mediterranean Sea (marked by the box with a dashed black frame in

Fig. 4), the $AOD_{nitrate}$ increases by ~0.01 and the cooling effect of nitrate is amplified by ~0.54 W m$^{-2}$


(negligible without introducing sea-salt aerosol). Sensitivity study also shows a monotonic increase of

$AOD_{nitrate}$ with $[Na^+]$ over the Mediterranean Sea (Fig. 5b), indicating a dominant role of the 'mass-

enhancement effect' over regions with limited anthropogenic influence.

Thus, abundant sea-salt aerosol and pre-existing fine mode nitrate (as if no sea-salt aerosol influence in

the 'Case_SeasaltOff') favor an efficient 're-distribution effect' that reduces $AOD_{nitrate}$ and moderates nitrate

cooling. To generalize the net impact of sea-salt-induced 're-distribution effect' and 'mass-enhancement

effect' on anthropogenic nitrate cooling, we conducted statistical analysis of the molar ratio between fine

nitrate in the 'Case_SeasaltOff' and total sodium in the 'Case_SeasaltOn' (RNS = $[NO_3^-]/[Na^+]$ in mol mol$^{-1}$

$^1$) with relation to the percentage change of nitrate AOD (surrogate of $DRF_{nitrate}$) between the two cases, i.e.,

($AOD_{nitrate\ Case\_SeasaltOn}$ − $AOD_{nitrate\ Case\_SeasaltOff}$)/$AOD_{nitrate\ Case\_SeasaltOn}$. We found that the 're-distribution

effect' tends to be strong enough to overwhelm the 'mass-enhancement effect' when the RNS value is in the

range of about 1 to 30, corresponding to ~70% of the data points in the European domain, as detailed in

Supplementary Information (Fig. S6a and Section S4). Note that only the surface concentrations were used

for calculating the RNS, because fine mode ammonium nitrate is mostly limited in the surface layer due to

emissions of $NH_3$ are in surface layer. The sea-salt (or sodium) aloft during transport process (as discussed

in section 3.1) did not exert the 're-distribution effect' before being mixed down to surface layer, and should

not be included in the analysis. Therefore, we carried out a statistical analysis with surface nitrate and sodium

concentrations, to draw a more robust conclusion. When the 're-distribution effect' is sufficiently strong, the





net reduction of $AOD_{nitrate}$ may even counteract the AOD enhancement contributed by the additional sea-salt

aerosol loading and lead to an overall reduction of total AOD in many regions in Europe (Supplementary

Information Fig. S5b and Section S5).

**3.4 The 're-distribution effect' over Europe**

Due to the frequent interactions between sea-salt aerosol and polluted air masses (Myhre et al., 2006;Xu

and Penner, 2012), the moderation of nitrate cooling is expected to be common over Europe, where the lower

atmosphere is characterized by RNS values between 1 and 30 (Fig. 6). As discussed above (Fig. 5a and Fig.

S5), considerable moderation is expected over inland and Central Europe, where $[Na^+] > 2.5 \ \mu g \ m^{-3}$ is

observed frequently (Gustafsson and Franzén, 2000, also see Fig. 5c). Such moderation could be even more

relevant over coastal and continental outflow regions of North America and South/East Asia (discussed in

the next section), where high loadings of nitrate were observed and found to be significantly associated with

sea-salt aerosol (30-90% of total nitrate) (Xu and Penner, 2012).

To quantify the possible overestimation of nitrate radiative forcing when only the sea-salt-induced

'mass-enhancement effect' was treated but not the 're-distribution effect', similar statistical analysis (Fig. 6)

is conducted for the percentage change of nitrate AOD between the $AOD_{nitrate}$ in the 'Case_SeasaltOn' and

the corresponding $AOD^*_{nitrate}$, i.e. $(AOD_{nitrate} - AOD^*_{nitrate})/AOD_{nitrate}$. As described before, $AOD_{nitrate}$ stands

for the case where both 'mass-enhancement' and 're-distribution' effects have been accounted for the

'Case_SeasaltOn', while only the 'mass-enhancement effect' is accounted for $AOD^*_{nitrate}$. Fig. S6b shows

that the $AOD^*_{nitrate}$ could increase by 5-30% when introducing sea-salt and only the 'mass-enhancement

effect' is considered. Conversely, the statistics show that the 're-distribution effect' tends to significantly

reduce the nitrate AOD by 10-20% when there is abundant sea-salt interacting with anthropogenic nitrate

$(RNS < \sim 30$, Fig. 6). Note that this estimation of percentage reduction in $AOD_{nitrate}$ is robust (may be slightly

conservative), despite of the overestimation of particulate nitrate over Europe (Supplementary Information

Section S6). The upper limit of our result is comparable to a previous estimate of 25% reduction of $AOD_{nitrate}$

by sea-salt aerosol on a global scale by Myhre et al. (2006), where a similar 're-distribution effect' of sea-

salt aerosol was evaluated but the simplifications of the mass transfer between fine and coarse modes may

lead to overestimation of the reduction (Supplementary Information Section S1). The 're-distribution effect'

becomes rather weak (limited within 10%) as RNS further increased $(RNS > \sim 30)$ when marine air masses



are transported further inland.

### 3.5 The 're-distribution effect' over North America and on a global scale


An additional simulation over North America confirms our findings. Similar as over Europe, the sea-salt-induced 're-distribution effect' predominates and reduces $AOD_{nitrate}$ over industrialized and outflow regions of the North America domain (Fig. 7). Generally, similar reduction of $AOD_{nitrate}$ is found over North America due to the sea-salt-induced 're-distribution effect'. A monotonous decrease (increase) of column

fine (coarse) nitrate loading is observed all over North America and oceanic regions (Fig. 7c and Fig. 7d). This sea-salt-induced 're-distribution effect' overwhelmes the 'mass-enhancement effect' over most regions and thus reduces $AOD_{nitrate}$ (Fig. 7a) and cooling effect of nitrate especially over the Gulf of Mexico (Fig. 7b), although the total nitrate column loading increases significantly (Fig. S7). This is because the high concentration of nitrate in the continental outflow from North and Central America strongly interacts with

sea-salt over the Gulf (Xu and Penner, 2012).

The statistical analysis over North America shows a similar pattern as over Europe (Fig. 8). Meanwhile, different from Europe, there is large fraction of the model results presented in the RNS > 30 regime in North America, indicating less regions over North America experience a strong 're-distribution effect'. The oceanic influence dominates over western Europe whereas over the North America the more continental air masses

(Kottek et al., 2006) may be a possible reason of this. It leads to less interactions of sea-salt with anthropogenic nitrate in North America, and more significant reduction of $AOD_{nitrate}$ over Europe. This result is consistent with a previous study (Myhre et al., 2006). However, there are still chances that sea-salt transport (May et al., 2018) and impact $AOD_{nitrate}$ further inland over North America. On average, only considering the 'mass-enhancement effect' but ignoring the 're-distribution effect' may lead to an

overestimation of $AOD_{nitrate}$ by more than 20% when RNS < 1 and by about 10-20% when 1 < RNS < 30. This RNS range encompasses most of the coastal and offshore regions of North America and favors 're-distribution effect'.

On a global scale, the potential influence of the 're-distribution effect' on $DRF_{nitrate}$ is estimated by a one-year RNS simulation with the EMAC (Klingmüller et al., 2014;Pringle et al., 2010) chemistry-climate

model (Fig. 9). In line with the WRF-Chem results, a strong 're-distribution effect' is expected over North America and Europe, especially over the coastal regions with high nitrate loading and RNS values around



1. The coastal and offshore regions of Asia with $1 < RNS < 30$ may experience strong reductions of $AOD_{nitrate}$, where the 're-distribution effect' is expected to overwhelm the 'mass-enhancement effect'. The influence of sea-salt aerosol becomes negligible over inland Asia where marine air mass influence is small ($RNS > 30$). For the open sea regions with $RNS < 0.01$ (white background), nitrate climate effect is not important, due to very limited nitrate loading.

## 4. Summary and Discussion

The interaction between natural sea-salt aerosols and anthropogenic nitrate leads to the 're-distribution effect', which can shift the particulate nitrate from sub- to super-micron sizes and hence lower its mass light extinction efficiency and shorten its lifetime. This 're-distribution effect' can significantly moderate nitrate cooling.

The interaction between natural sea-salt aerosols and anthropogenic nitrate happens frequently over Europe (~90%). We performed a series of sensitivity studies during a typical sea-salt event over Europe, using WRF-Chem model with fully dynamic aerosol mass transfer treatment online-coupled. The 're-distribution effect' of nitrate is observed by field measurements and well captured by the 'Case_SeasaltOn' simulation. Over the highly polluted northern Poland region, our sensitivity modelling results show that the 're-distribution effect' can reduce $AOD_{nitrate}$ by about 20-30%, or even up to ~50% when the sea-salt event is sufficiently strong. Conversely, if only consider the increase of nitrate mass by introducing sea-salt ('mass-enhancement effect') and ignore the 're-distribution effect', nitrate AOD could increase by ~8% or even up to ~30% in a strong sea-salt event. We propose a single parameter, RNS ($[NO_3^-]/[Na^+]$ in mol mol$^{-1}$), to describe the competition between sea-salt-induced 're-distribution effect' and 'mass-enhancement effect'. In general, (1) the sea-salt-induced 'mass-enhancement effect' is dominant over oceanic regions and tends to increase $AOD_{nitrate}$ when RNS is lower than 1; (2) the sea-salt-induced 're-distribution effect' on nitrate PMSD can decrease $AOD_{nitrate}$ by about 10-20% and overwhelm the 'mass-enhancement effect' resulting in a net reduction of $AOD_{nitrate}$ when $1 < RNS < 30$; and (3) the influence of sea-salt aerosol is not significant when RNS is higher than 30. These findings are further confirmed by the sensitivity simulations over North America. The impact of 're-distribution effect' on a global scale is estimated using global simulation of RNS, as shown in Fig. 9. Significant 're-distribution effect' is expected over Europe, the Gulf of Mexico, coastal and offshore regions world-wide, may resulting in reduction of $AOD_{nitrate}$ by about 10-20%.



## 5. Implication

This study highlights the impact of the 're-distribution effect' on moderating nitrate cooling by interacting with natural sea-salt aerosols. A similar 're-distribution effect' may apply to the heterogeneous

reaction and consumption of gaseous organic compounds, sulfuric and nitric acids on natural desert dust (Usher et al., 2003;Ponczek and George, 2018;Dupart et al., 2012;Ravishankara, 1997), although non-volatile particulate sulfate does not shift from fine to the coarse mode aerosols. Uptake of acids by dust particles can shorten their lifetimes and reduce their radiative forcing (Liao and Seinfeld, 2005;Harris et al., 2013;Karydis et al., 2016;Abdelkader et al., 2015), which could be significant over inland areas where sea-

salt aerosol is lacking. Our results imply the possibility that natural particles (sea-salt aerosol and very likely dust as well) moderate the DRF of anthropogenic aerosols and alter the nitrogen (very likely sulfur also) deposition efficiency (Fig. 1). More comprehensive modelling studies with fully dynamic aerosol mass transfer treatment are needed for improving the assessment of aerosol climate effect accounting for the 're-distribution effect' on a global scale.




## Acknowledgments

We would like to acknowledge the following funding resources: National Natural Science Foundation

of China (91644218), the National Key R&D Program of China (2017YFC0210104) and Guangdong

Innovative and Entrepreneurial Research Team Program (2016ZT06N263). We thank Dr. Konrad Müller

(TROPOS) for his contribution to the aerosol composition measurements. This work was supported by

German Research Ministry (01LK1212C) and the Max Planck Society (MPG). Y. F. C. would also like to

thank the Minerva Program of MPG.

## Author contributions

Y. F. C. led the study. Y. F. C. and Y. C. conceived and design the study. Y. C. performed the WRF-

Chem model simulations and processed the data. N. M. supported the optical calculation. C. W. supported

the kinetic part of the model simulation and result analyses. A. P. and J. L. provided the EMAC global

simulation. G. S. carried out the aerosol chemical composition observations at Melpitz. Y. C., Y. F. C. and

H. S. interpreted the results. All co-authors discussed the results. Y. C. and Y. F. C. wrote the manuscript with

inputs from all co-authors.

## Additional information

Correspondence and requests for materials should be addressed to and Y. F. C and Y. C.

## Competing financial interests

The authors declare no competing financial interests.

## Data availability

WRF-Chem model code and FINN fire emissions are openly available for download from the website

www2.acom.ucar.edu. NCEP reanalysis data is openly available for download from the website

https://rda.ucar.edu/. AERONET and aerosol observational datasets are openly available for download from

the websites https://aeronet.gsfc.nasa.gov/ and http://ebas.nilu.no/default.aspx. The European emission

inventory is available from AQMEII project (http://aqmeii.jrc.ec.europa.eu/) and EUCAARI project

(https://www.atm.helsinki.fi/eucaari/?q=node/3). The global emission inventory is available from EDGAR

project (http://edgar.jrc.ec.europa.eu). The results of EMAC global model is available from

https://dx.doi.org/10.17635/lancaster/researchdata/297. All data needed to evaluate the conclusions in the

paper are present in the paper and/or the Supporting Information. Additional data related to this paper should

be addressed to and Y.F.C and Y.C.



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



**Table 1**. Configurations of WRF-Chem

| Physics | WRF options |
|---|---|
| Micro physics | Lin scheme (Lin, 1983) |
| Boundary layer | YSU (Hong, 2006) |
| Surface | Rapid Update Cycle (RUC) land surface model |
| Shortwave radiation | Goddard shortwave (Chou et al., 1998) |
| Longwave radiation | New Goddard scheme |
| Cumulus | Grell 3D |
| Urban | 3-category UCM |
| **Chemistry and Aerosol** | **Chem options** |
| Aerosol module | MOSAIC with 8 bins (Zaveri et al., 2008) |
| Gas-phase mechanism | CBMZ (Zaveri and Peters, 1999) |
| Photolytic rate | Fast-J photolysis scheme (Wild et al., 2000) |
| Sea salt emission | Gong scheme (Gong, 2003) |





**Fig. 1. Concept of the sea-salt aerosol induced 're-distribution effect'.**

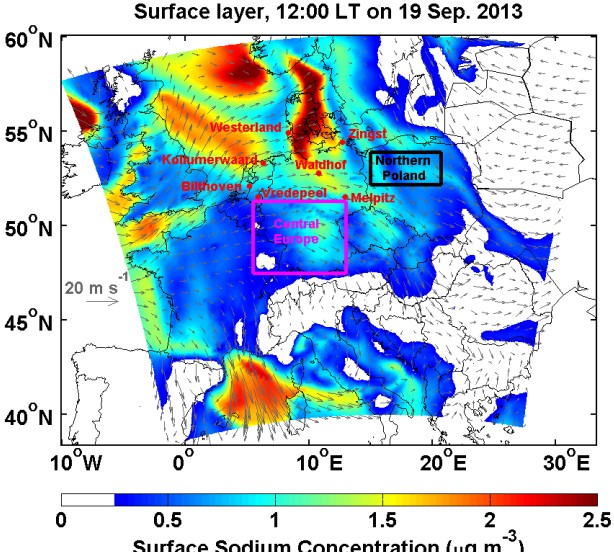

**Fig. 2. Surface sodium mass concentration over Europe domain at 12:00 local time (LT) on 19 September 2013.** The 10-meter wind is indicated by the grey arrows. The results are based on the 'Case_SeasaltOn'. Westerland, Waldhof, Zingst, Bilthoven, Kollumerwaard, Vredepeel, Melpitz, Central Europe and the northern Poland regions are marked.

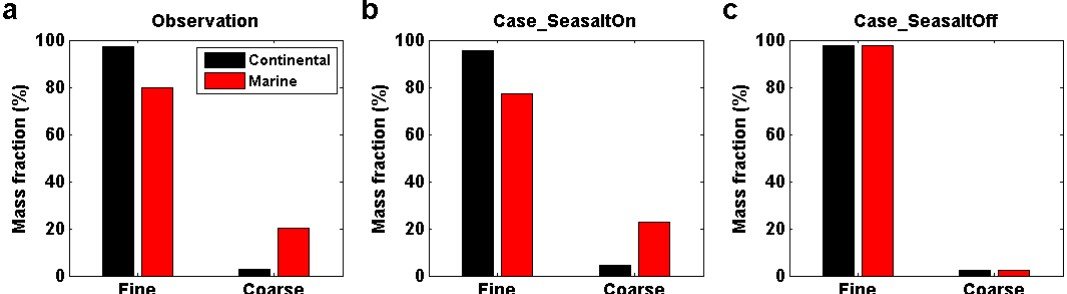

**Fig. 3. Observed and simulated mass fraction of particulate nitrate in fine and coarse modes at Melpitz, Germany. a**, Observation based on the Berner impactor measurements. **b**, WRF-Chem 'Case_SeasaltOn' simulation, i.e., with sea-salt emission. **c**, WRF-Chem 'Case_SeasaltOff' simulation, i.e., without sea-salt emission. The results are grouped into continental (black bar) and marine (red bar) air mass types, respectively. According to the size-cuts of the Berner impactor, the size ranges of the fine and coarse mode particles are defined as $PM_{1.2}$ (particles with an aerodynamic diameter smaller than 1.2 $\mu$m) and $PM_{1.2-10}$ (particles with an aerodynamic diameter larger than 1.2 $\mu$m and smaller than 10 $\mu$m), respectively.



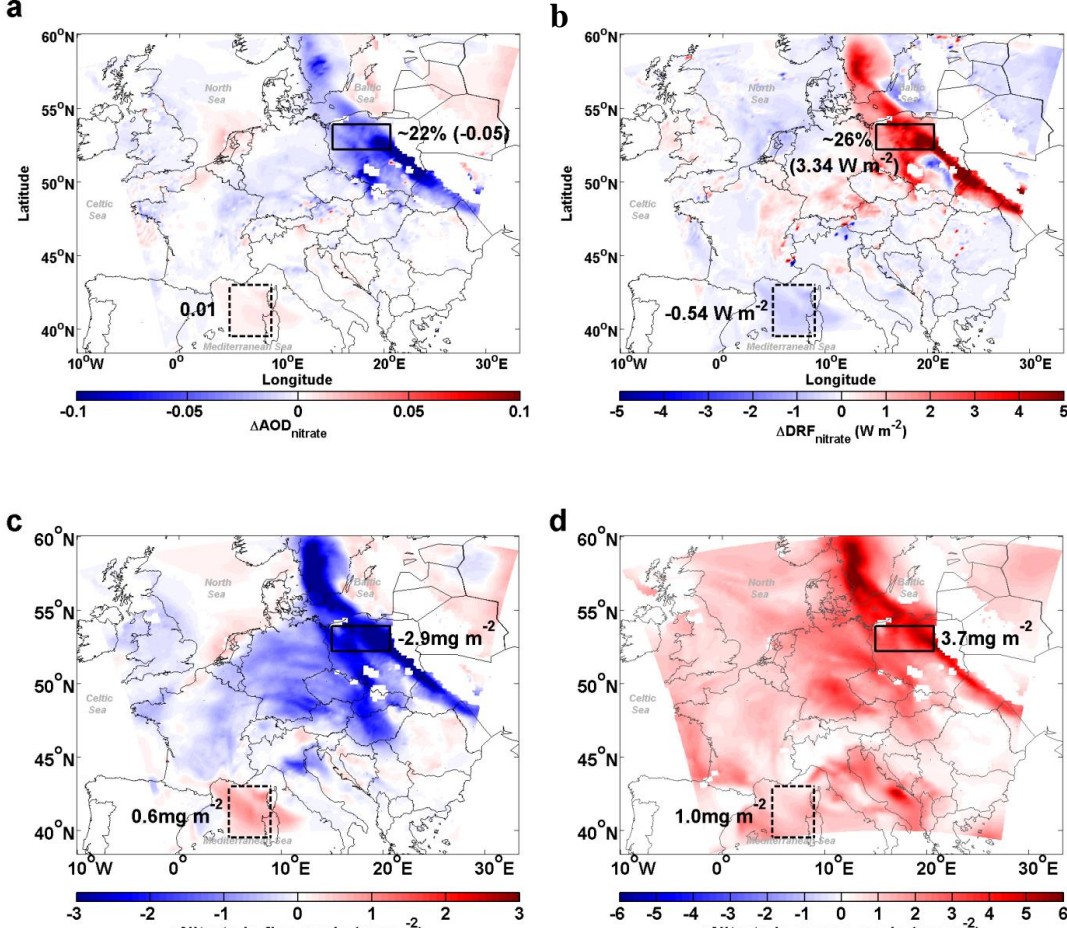

**Fig. 4. Influence of sea-salt on the abundance and direct radiative forcing of nitrate.** Differences ('Case_SeasaltOn' – 'Case_SeasaltOff') between simulations with and without sea-salt emission in aerosol optical depth ($\Delta AOD_{nitrate}$, **a**), direct radiative forcing ($\Delta DRF_{nitrate}$, **b**), and column loading of nitrate ($\Delta Nitrate$) for the fine (**c**) and coarse (**d**) mode particles, during daytime, i.e., 07:00-16:00 local time (LT) on 19 September 2013. The northern Poland and Mediterranean regions are marked by boxes with solid and dashed black frames, respectively.

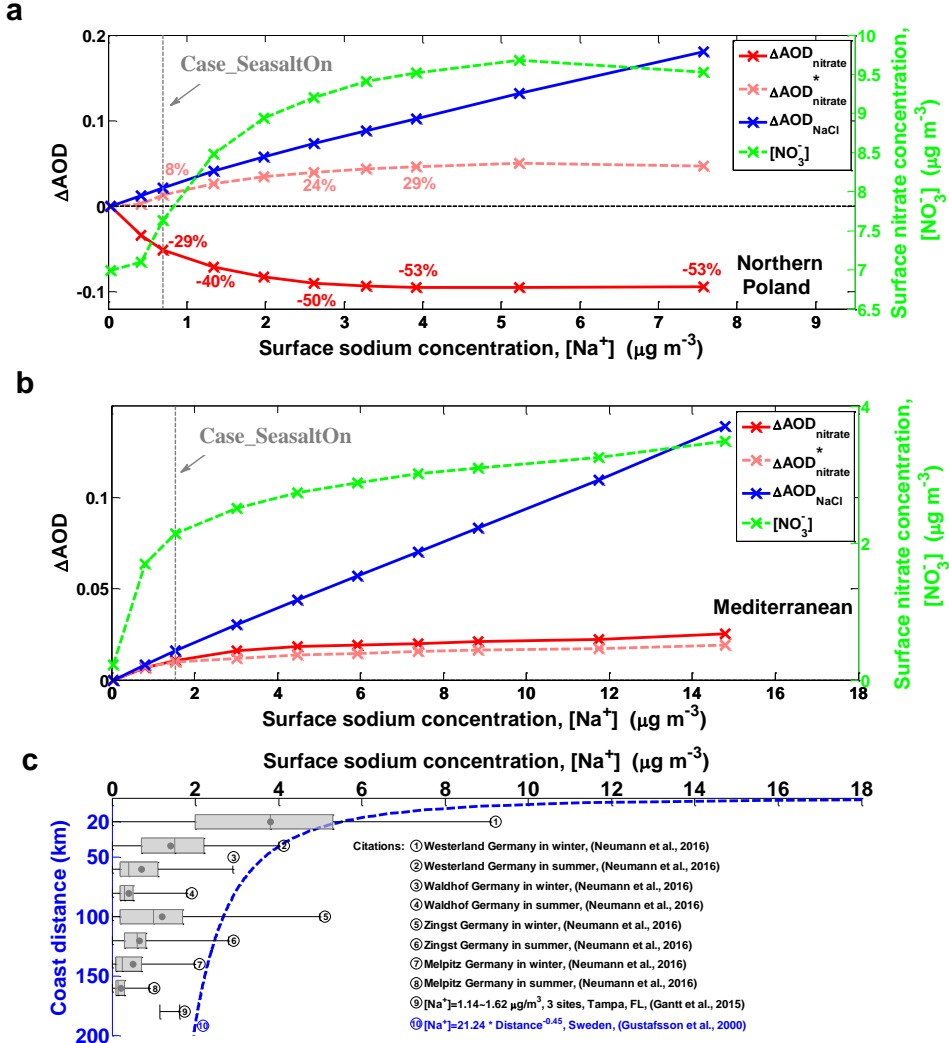

**Fig. 5. Sensitivity of aerosol optical depth ($\Delta$AOD) as a function of sodium mass concentration ([Na$^+$]).** Differences between the 'Case_SeasaltOff' and sensitivity cases (sea-salt emission with different scaling factors) for different aerosol components, i.e., nitrate ($\Delta$AOD$_{nitrate}$) and sea-salt ($\Delta$AOD$_{NaCl}$): $\Delta$AOD (sensitivity case – 'Case_SeasaltOff') versus [Na$^+$] over northern Poland (**a**) and Mediterranean regions (**b**), respectively. The model results shown here are averages over a sea-salt event during daytime (07:00-16:00 LT) on 19 September 2013. The results of surface [NO$_3^-$] are indicated by the green dashed line. Here, the 'Case_SeasaltOff' is the reference case and the 'Case_SeasaltOn' is marked. Note: $\Delta$AOD$^*_{nitrate}$ (pink dashed line) indicates the $\Delta$AOD of nitrate calculated by re-allocating nitrate mass into different size bins according to the normalized nitrate particle mass size distribution simulated in 'Case_SeasaltOff' (i.e., without 're-distribution effect'). **c,** The measured [Na$^+$] at different sites over Sweden as a function of coast distance, and in Germany (marked in Fig. 2) and US. The box-whisker plots of the references ①-⑧ indicate the median, mean (black dot), 25% percentile, 75% percentile, maximum and minimum. The error bar of the reference ⑨ indicates the range. The blue dashed line of the reference ⑩ indicates the statistically empirical function of [Na$^+$] with the distance from coast, based on the network measurements of 16 sites in Sweden.





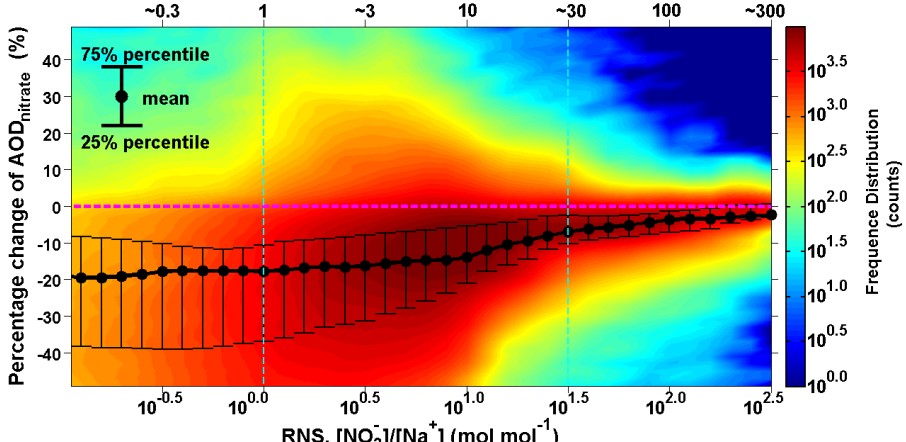

**Fig. 6. Intensity of 're-distribution effect' as a function of molar ratio between surface fine nitrate and total sodium (RNS) over European domain.** The intensity of the re-distribution effect is calculated as the difference in percentage between $AOD_{nitrate}$ and $AOD^*_{nitrate}$ in the 'Case_SeasaltOn'. The black dots indicate the mean values; the upper and lower error bars indicate the 75% and 25% percentile, respectively. The colour indicates the frequency distribution (i.e., how many counts) of the hourly model results over entire Europe domain during 16-20 September 2013. The coverage of model results between the two light blue dashed lines is ~70%.



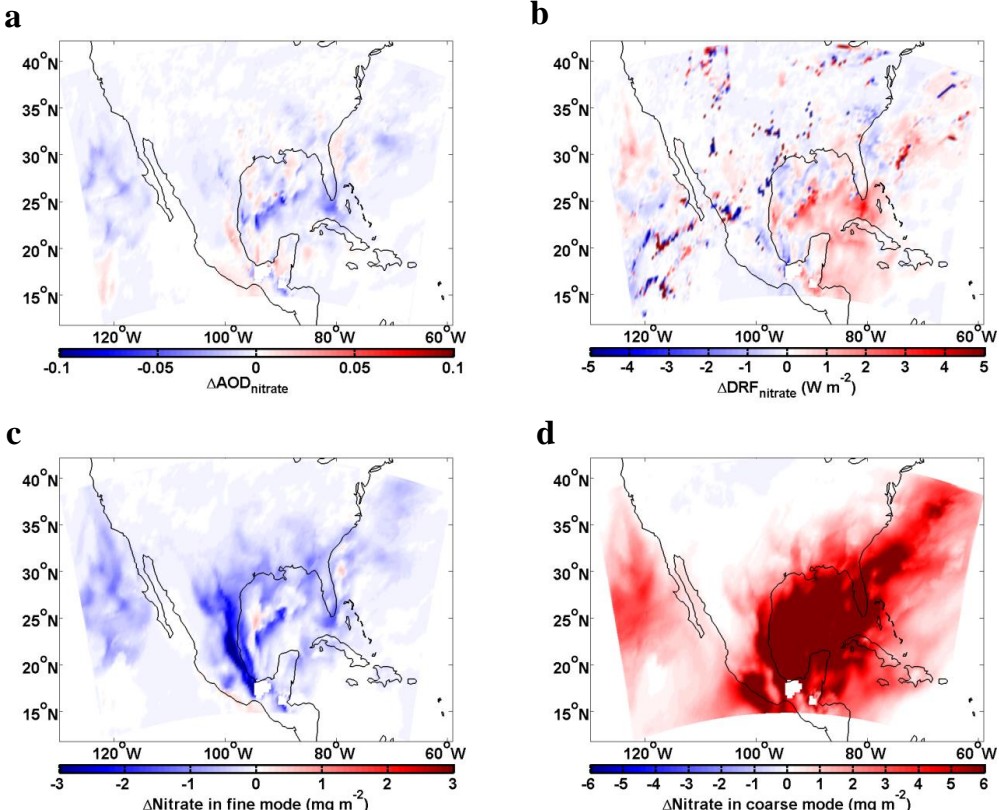

**Fig. 7. Influence of sea-salt aerosol on the abundance and direct radiative forcing of nitrate (similar as Fig. 4).** Differences ('Case_SeasaltOn' – 'Case_SeasaltOff') between simulations with and without sea-salt emission in aerosol optical depth ($\Delta AOD_{nitrate}$, **a**), direct radiative forcing ($\Delta DRF_{nitrate}$, **b**), and column loading of nitrate ($\Delta Nitrate$) for the fine (**c**) and coarse (**d**) mode particles. The modelled results showed here are averaged during daytime over North America domain, i.e., 16:00-22:00 (UTC) on 10-17 January 2015.





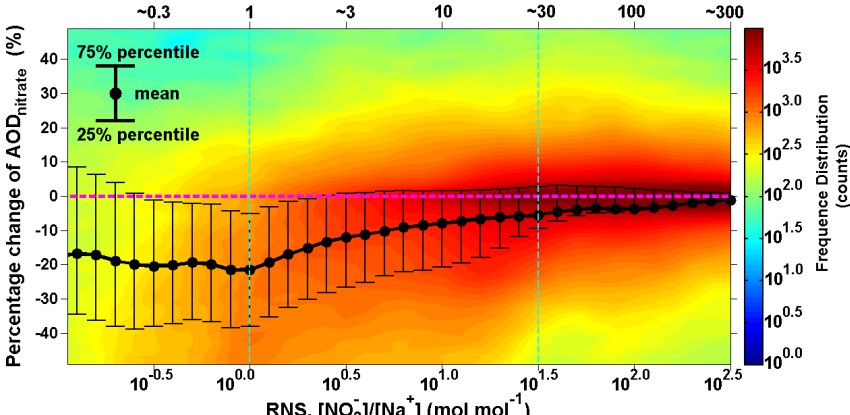

**Fig. 8. Intensity of 're-distribution effect' as a function of molar ratio between surface fine nitrate and total sodium (RNS) over North American domain, similar as Fig. 6.** The intensity of 're-distribution effect' is calculated as the difference in percentage between $AOD_{nitrate}$ and $AOD^{*}_{nitrate}$ in the 'Case_SeasaltOn'. $AOD^{*}_{nitrate}$ indicates the $AOD_{nitrate}$ calculated by re-allocating nitrate mass into different size bins according to the normalized nitrate particle mass size distribution simulated in 'Case_SeasaltOff' (i.e., without 're-distribution effect'). The black dots indicate the mean values; the upper and lower error bars indicate the 75% and 25% percentile, respectively. The colour indicates the frequency distribution (i.e., how many counts) of the hourly model results over entire North America domain during 10-17 January 2015.

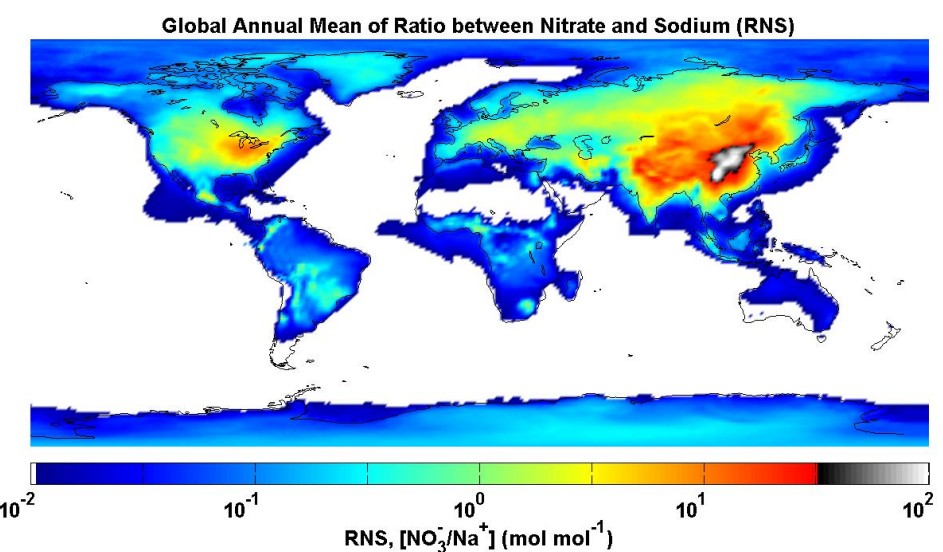

**Fig. 9. The global distribution of annual mean RNS, simulated with the EMAC model.** The regions with RNS smaller than 0.01, i.e., negligible particulate nitrate loading, are white.