# Peer review of "Natural sea-salt emissions moderate the climate forcing of anthropogenic nitrate"

_Atmospheric Chemistry and Physics, 2019_

## Referee Comment (RC1) · Anonymous Referee #2 · 6 Sep 2019

This is a well-written paper that highlights an interesting topic that I had not previously thought about, but makes a lot of sense conceptually and could have important implications for anthropogenic aerosol radiative forcing on a global scale. I have one comment which questions some of the methods used in the experiment, which I am not expecting the authors to address in this paper but could be the subject of further discussion or future work. Beyond that, I just have a few minor comments I think the authors should

My main comment is that the main purpose of this paper is to investigate the sensitivity of the nitrate aerosol radiative forcing to the process that redistributes it from fine modes when associated with ammonium to course modes when associated with sodium. The authors have tested this by turning off sea salt emissions in the sensitivity simulation and seeing how this changes nitrate aerosol size distribution, which

is perfectly valid but does limit what they can investigate in some ways because removing sea salt emissions will have many other changes to total aerosol AOD, CCN and aerosol-radiation/aerosol-cloud interactions that go beyond the redistribution effect they are investigating. I would have thought a better experiment would have been to turn off the association between NO3 and Na in MOSAIC in the sensitivity simulation, and have sea-salt emissions in both, thereby having a "clean" experiment where the only change is to the exact process you are investigating. A key benefit of this approach is that you would then be able to test the impact of this process on both total aerosol AOD/DRF (not just the NO component), and how it effects the aerosol-cloud interactions (ACI).

On both a local and global scale, my gut says that the influence of the redistribution effect on (ACI), particularly the Twomey or first-indirect effect, would be greater that the DRF. I would expect shifting nitrate from subnitrate modes to coarse modes would slightly increase the number of low-supersaturation CCN, whilst massively decreasing the number of high-supersaturation CCN. This could in turn have a large impact on the activated cloud droplet number, cloud optical depth and therefore climate. Unfortunately, the current methodology cannot address this as any impact from the redistribution effect would be likely overwhelmed by the lack of sea salt aerosol in the sensitivity simulation. This is a shame, as without investigating the aerosol-cloud interactions we are only seeing part of the impact of this process on the climate.

As I said above, I'm not expecting the authors to do reruns with this change as it is a substantial amount of work and, to my knowledge, their current paper is already the first I know of to look into this problem in detail and already contributes significantly to the field. But I would be interested in hearing their response and any follow up studies that look into this further – this discussion could be included in the revised paper.

Minor comments: Ln 40-41: Under what kinds of emission scenarios are we expecting the aerosol radiative cooling to increase by this much?

Ln 76: Please explain acronym for HOPE-Melpitz campaign Ln 82: "which represent continental period and marine period" Is this based from back trajectories? Please explain better.

Section 2.2 Model Description: this section is missing some important information. Firstly, I couldn't find anywhere a description of how the MOSAIC mechanism describes the chemical uptake of HNO3 onto aerosol via NH4+ or Na+ channels. A short (one-two sentences) here describing that process and how it is parameterised in MOSAIC would be welcome.

Please can you add what meteorology data you are using to drive the model and if you are nudging, or if it freerunning (with or without feedbacks). Some of this information is in the supplement.

Finally, are you running with N2O5 heterogeneous chemistry? This is relevant for a point later on in the paper, plus in WRF-Chem the Bertram and Thornton (2009) parameterisation does have a channel for uptake via Cl-, so provides another mechanism for nitrate to be taken up to course modes that is modulated by the presence of sea salt (Archer-Nicholls et al., 2014).

Bertram, T. H., & Thornton, J. A. (2009). Toward a general parameterization of N2O5 reactivity on aqueous particles: the competing effects of particle liquid water, nitrate and chloride. Atmospheric Chemistry and Physics, 9(4), 15181–15214. https://doi.org/doi:10.5194/acp-9-8351-2009

Archer-Nicholls, S., Lowe, D., Utembe, S., Allan, J., Zaveri, R. A., Fast, J. D., et al. (2014). Gaseous chemistry and aerosol mechanism developments for version 3.5.1 of the online regional model, WRF-Chem. Geoscientific Model Development, 7, 2557–2579. https://doi.org/10.5194/gmd-7-2557-2014

Ln 110: Is the version of MOSAIC being used with or without aqueous chemistry?

Ln 137: Please include figure of outer and inner domains, here or in supplement.

Ln 249: In terms of lifetime of nitrate radical (i.e. gas phase NO3), I would have thought this has more to do with the uptake of N2O5 to aerosol enhanced by Cl- rather than the HNO3 process you discuss in this paper. I'm not sure if it is quite relevant with the discussion here, which is more about the lifetime of aerosol NO3- when it is in different aerosol size modes. Please clarify.

Ln 336-347: Would it be possible to show a map of RNS over Europe and North America from the WRF-Chem model for comparison? You've only done this for the global EMAC model, which by the sounds of things has lower values for RNS over most of the US compared to WRF-Chem (essentially nowhere in North America has RNS > 30 in the EMAC model, whereas this regime dominate continental North America in the WRf-Chem simulations). The figures currently given make it hard to follow the logic in this paragraph. In general, I think this section can be improved by outlining the conclusions that can be drawn from the evidence given, rather than speculating on what we might expect the effects to be if sea salt is transported further inland.

Ln 348-350: Would it be possible to plot a first-order estimate of what the change to nitrate AOD would be from this effect, using a function based on the relation between RNS and [NO3-] derived from the WRF-Chem model simulations?

Ln 364: "($\sim$90%)" – what is this referring to exactly?

Ln 390-394: This last section is confusing what you can derive from the results of this study (modulation to nitrate DRF from sea-salt aerosol) from that speculated from other related chemical interactions (sulfur, dust etc.). Please rewrite to clearly separate the conclusions drawn from your results from speculations/highlight topics for future research.

Figures:

Please promote Fig S1 into the main paper, I think it really clearly shows how the redistribution effect changes the aerosol chemical size distribution.

Fig 9. Please can you include a subplot with this showing [NO3-]? You need this with RNS to get an understanding of where the redistribution impact is likely to be strongest.

[Figure]

---

## Referee Comment (RC2) · Daniel Neumann (Referee) · 9 Sep 2019

The submitted manuscript describes the impact of sea salt aerosol on atmospheric nitrate concentrations. The presence of sea salt aerosol leads to a mass-enhancement of nitrate in the particle phase but also to a re-distribution of nitrate from fine to coarse particles modes. However, the two processes increase and decrease the absolute value of nitrate direct radiative forcing, respectively, and, hence, are in competition to each other. In theory, the considered impact of particulate sea salt on the size distribution of particulate nitrate and the resulting change of the DRF is well understood. Practically, however, we do not have exact numbers on how strong the impact is. The authors evaluate this competition for certain environmental conditions and spatial scales.

[Figure]

The manuscript is well written and structured. The scientific aim is clearly formulated. The main manuscript provides the relevant information as brief as possibles, whereas an extended presentation of the model description and results are provided in the supplement. I appreciate the quite detailed supplementary text and the additional figures.

I have several comments on the manuscript. However, these are no critical aspects. In summary, I recommend the manuscript for acceptance after minor revision.

**1  Major comments**

What are the statistics of the sea-salt-over-continent-transpor events in Europe? Do they take place sufficiently often and/or do they persist over a sufficient long time period in order to have relevant impact on the annual mean $DRF_{nitrate}$?

For the European modeling period a time frame containing a strong sea-salt-transport event was choosen. For the US example, an arbitrary time period was choosen. How far are both modeling cases – Europe and US – comparable?

On p.4 l.116 the authors state that they found in previous studies that – using the Gong emission parameterization – modeled sea-salt concentrations overestimated measurements by a factor of 10. However, the sea salt emissions estimated by the Gong parameterization are not necessarily 10x as high as the real world sea salt emissions are. Processes that lead to lower atmospheric sea salt concentrations might be underestimated in the used model – e.g. underestimated deposition. Or, the particle size distribution of the Gong sea salt emissions might not be appropriate for the model setup. Therefore, please add "... using WRF-Chem ..." or something similar to the sentence – e.g. in the end of the sentence.

In the first paragraph on page S9 in the supplement, the authors discuss that nitrate concentrations are overestimated by several models in Europe. Vivanco et

**[ACPD](ACPD)**

Interactive
comment

al. (2017, DOI: https://doi.org/10.1016/j.atmosenv.2016.11.042) evaluated several air quality models and found that most considered models underestimate the nitrogen deposition in Europe (The EMEP model performed best). WRF-Chem was not partof the latter evaluation. However, too high atmospheric nitrate concentrations may point to issues in the deposition parameterizations. This wouold also explain why the Gong parameterization apparently lead to considerably overestimated atmospheric sea salt concentrations. Could the authors comment on that?

Evaluating the mass concentration of sea salt aerosol is reasonable to evaluate the coarse sea salt emissions. The contribution of fine sea salt to the total sea salt mass concentration is very low. Hence, the strong overestimation of sea salt mass concentrations documented in previos studies does not necessarily indicate that the fine particulate sea salt was overestimated as well. The black bars in Fig. S3 rather indicates that fine sea salt might be underestimated in some episodes. Please briefly describe this uncertainty in the Materials & Methods or in the Discussion section.

p.6 l.165-168 "In addition, a one-year simulation with global model (EMAC) was carried out for analysis of the potential impact of 're-distribution effect' on a global scale, although the fully dynamic mass transfer between particle sizes is not considered in EMAC (four size modes rather than eight size bins as applied in the WRF-Chem model)." Why is EMAC neverthelss applicable for the used purpose?

"Sea-salt is emitted into the marine planetary boundary layer (PBL) as coarse particles [. . . ]" (p.7 l.183). Aren't it coarse and fine particles? Depending on the considered moment of the particle size distribution the fine or the coarse mode dominates: the number distribution is dominated by the fine mode; the volume/mass distribution is dominated by the coarse mode. Please clarify this in the text.

The authors mention the importance of vertical transport of sea salt aerosol into the continental free troposphere followed by the horizontal transport of it over land and refer to their previous studies (p.7 l.186-190). This transport mechanism seems to be very

important. However, vertical transport/convection is not necessarily well represented by all meteorological models – depending on grid resolution and parameterization. This study's WRF-Chem model simulations probably reproduce this vertical transport of sea salt? Do the other used models reproduce it as well?

In chapter 3.3 (p.10 l.281-290), the authors mention the aim to 'generalize' the results. To 'generalize' something has a quite broad meaning and can be interpreted differently. The benefit of calculating the RNS and of including Fig. 5 did not become clear for me in chapter 3.3 – but, later in chapter 3.4 it became obvious. Please clarify the motivation in chapter 3.3.

Classification of RNS: Currently, we have the situations "$RNS < 1$", "$1 < RNS < 30$" and "$30 < RNS$". I know that these are rough classifications. Nevertheless, please also cover "$RNS == 1$" and "$RNS == 30$" to be mathematically correct.

**2   Comments on figures**

Figure 1: I like the intention of Fig. 1. All relevant aspects are shown. However, on the first view, the reader might not recognize the particle size distribution plot as such. At least I did not recognize it first. I am not sure how to improve the Figure. Possibly, a y-axis on the left of the plot might help. Their are no issues when the figure is printed in grey-scales (see my comments to the other figures below).

Figure 2, 6, 8 and 9: Please consider using a color scale, which is recognizable in black-and-white and readable by color blind people (not "jet" or "rainbow"). Examples for such a color scale are "viridis" and "magma".

Figure 4 and 7: Printed in grey scales it is hard to distinguish whether values are negative or positive. However, alternative color scales would make the full-color plots difficult to interpret. Therefore, it might be reasonable to keep the blue-white-red

colorscale.

Figures 3: Please add information to the plot that nitrate is plotted. It is written in the caption but it would be nice to also have this information in the plots. The colors in this figure are fine.

Figures 5: The colors in this figure are fine.

Figure S6: Maybe exchange panels (a) and (b). For me it would have made the figure easier to understand. I would consider adding Figure S6 to the main manuscript (and combine it with Fig. 6) because it adds valueable information to the chapter 3.4.

**3 Minor comments**

p.4 l.117: consider to replace "for both with and without sea-salt presence respectively" by "for both sea salt emission cases"

p.5 l.123: remove "and" in "... assumption and taking ..."

p.5 l.136: "and part of North Africa" → "and the northern part of North Africa"

p.5 l.142 "$1/8°\ \times 1/16°$" → space between "×" and "1/16"

p.5 l.147 "In the 'Case SeasaltOn' (with sea-salt emission) of European simulation" → "In the European 'Case SeasaltOn' simulation (with sea-salt emission)", suggestion

p.5 l.148 "over coastal region" → "over coastal regions", added plural-s

p.6 l.149/150 "with a factor (and correlation coefficient) of 0.85 (0.67), 1.16 (0.80) and 0.83 (0.87) respectively for Bilthoven, Kollumerwaard and Vredepeel (Fig. S2)" → "with a factor (and correlation coefficient) of 0.85 (0.67), 1.16 (0.80) and 0.83 (0.87) for Bilthoven, Kollumerwaard and Vredepeel, respectively (Fig. S2)", moved 'respectively'

p.6 l.154 "over coastal, German low lands (Melpitz) and northern Poland regions" →

"over coastal, German low land (Melpitz) and northern Polish regions", remove 's' from lands because we have "coastal regions", "German low land regions" and northern Polish regions"

p.6 l.162 "concentration of nitrate was" → "concentrations of nitrate were"

p.6 l.164 "covers US, the Gulf of Mexico and part of Pacific and Atlantic oceans" → "covers the US, the Gulf of Mexico and parts of Pacific and Atlantic oceans", 'the' in front of 'US' and 's' added to 'part'

p.6 l.171 "1.1 by 1.1 degrees" → "$1.1° \times 1.1°$", formatting consistent with p.5 l.142

p.6 l.161 "boarder" → "broader"

p.6 l.177 "approach Central Europe, and the interaction" → "approach Central Europe. The interaction"

p.7 l.179 "a typical sea-salt transport event"; maybe 'sea salt transport event' in italics

p.7 l.182 "originated" → "originating"

p.7 l.187 "long-range transport, see the figure 11 of Chen et al. (2016a)" → "long-range transport (see Fig. 11 of Chen et al. (2016a))"

p.8 l.212: "the northern Poland region" → "a region in northern Poland"

p.8 l.220: ",thus " → ", thus, "

p.8 l.227: "Over the northern Poland region" → "Over the region in northern Poland"

p.8 l.231-234: "being slightly . . . in the offline calculation" – Please consider writing this sentence outside of the brackets. It improves the readability.

p.9 l.241-242: "with increasing AOD with [NO3-]" → "of increasing AOD with increasing [NO3-]"; I am not sure about it but for me it sounds better.

p.9 l.242: "would result by neglecting the 're-distribution effect'" → "would result when

the 're-distribution effect' was neglected"

p.9 l.242-244: "For example, instead of a decrease by 29

p.9 l.245: remove "also"

p.9 l.245: ". . . increases and there is . . . " → ". . . increases. There is . . . "

p.9 l.248: "and thus shorter" → "and, thus, shorter"

p.9 l.251: "and hence further moderate" → "further moderating"

p.9 l.254: ", therefore fasten" → "fastening"

p.9 l.255: "oxygenated nitrogen" → maybe "oxidized nitrogen"?

p.9 l.257: "which result" → "which resulted"

p.9 l.258-263: This has been mentioned previously. Although, this is important motivation for performing another simulation without deposition, I would like to suggest shortening these sentences considerably.

p.10 l.265-266: "as the simulations with aerosol dry deposition" → "as in the simulations with aerosol dry deposition turned on"; corresponds better with l.263-264 stating ". . . with aerosol dry deposition turned off"

p.10 l.289: "due to" → "because"; 'due to' does not allow a verb ('are')

p.11 l.319-320: "Myhre et al. (2006), where a similar . . . aerosol was evaluated but the simplification . . . " → "Myhre et al. (2006). They evaluated a similar . . . aerosol. But, their simplification . . . "; split into two or three individual sentences;

p.12 l.332: "and thus reduces" → "and, thus, reduces"

p.12 l.337: "presented" → "present"

p.12 l.342-343: "that sea-salt transport (May et al., 2018) and impact $AOD_{nitrate}$ further inland over North America." → "that sea-salt is transported further inland over North

America und impacts AOD$_\text{nitrate}$ there (May et al., 2018)."

p.13 l.352-356: Please refer to Fig. 9 (maybe I overlooked the reference).

p.13 l.360: "and hence lower" → "and, hence, lower"; I would consider splitting the sentence into two sentences at this 'and'.

p.13 l.369: "if only consider" → "if we only consider" or passive form

p.13 l.374: "...when RNS is lower than 1" → Formulation is ambiguous in this context: the dominance could take place when $\text{RNS} < 1$ OR the mass-enhancement effect could be associated with $\text{RNS} < 1$. It know (and probably most readers) know that the latter is meant. However, I would suggest to reformulate this part. The same for the next sentence ("...when $1 < \text{RNS} < 30$").

---

## Author Comment (AC1) · 16 Nov 2019

**Response to comments of referee #1: Dr. Daniel Neumann**

**General comments**

The submitted manuscript describes the impact of sea salt aerosol on atmospheric nitrate concentrations. The presence of sea salt aerosol leads to a mass-enhancement of nitrate in the particle phase but also to a re-distribution of nitrate from fine to coarse particles modes. However, the two processes increase and decrease the absolute value of nitrate direct radiative forcing, respectively, and, hence, are in competition to each other. In theory, the considered impact of particulate sea salt on the size distribution of particulate nitrate and the resulting change of the DRF is well understood. Practically, however, we do not have exact numbers on how strong the impact is. The authors evaluate this competition for certain environmental conditions and spatial scales. The manuscript is well written and structured. The scientific aim is clearly formulated. The main manuscript provides the relevant information as brief as possibles, whereas an extended presentation of the model description and results are provided in the supplement. I appreciate the quite detailed supplementary text and the additional figures. I have several comments on the manuscript. However, these are no critical aspects. In summary, I recommend the manuscript for acceptance after minor revision.

*Many thanks to Dr. Neumann for the careful reading and the helpful comments and suggestions. We have improved the manuscript accordingly. Please find a point-by-point response below. Please refer the order of figures to the revised version.*

**Major comments:**

1) What are the statistics of the sea-salt-over-continent-transpor events in Europe? Do they take place sufficiently often and/or do they persist over a sufficient long time period in order to have relevant impact on the annual mean DRF$_{nitrate}$?

*Marine air masses influence Europe quite frequently. In a previous long-term observation-based study (Birmili et al., 2001), they classified the air masses influencing Central Europe as 'Maritime Character', 'Continental Character' and 'Mixed-Type Character'. They performed a statistical study with observations of more than one year and reported that the frequency of*

*'Maritime Character' and 'Mixed-Type Character' air masses are 46.6% and 33.3%, respectively (see the Table 1b of Birmili et al. 2001). Therefore, the marine air masses influence Europe with a total frequency of about 90%, and are expected to have impact on the annual mean DRF_nitrate by such a high frequency. We have modified the summarization of the frequency in section 3.1 to make it clearer, as shown below.*

*"Marine air masses frequently (~90%) approach Central Europe (Birmili et al., 2001). The interaction between anthropogenic pollutants and sea-salt aerosol commonly happens in the atmosphere."*

***Changed to:***

*"Marine air masses frequently, up to 90% of days in a year, influence Central Europe (Birmili et al., 2001). The interaction between anthropogenic pollutants and sea-salt aerosol commonly happens in the atmosphere."*

2) For the European modeling period a time frame containing a strong sea-salt-transport event was choosen. For the US example, an arbitrary time period was choosen. How far are both modeling cases – Europe and US – comparable?

*There are similarities and differences between the European case and the US case.*

*The difference. For Europe case, a sea-salt event was chosen, we see a strong 're-distribution effect'. This leads to a reduction of 10-20% in AOD_nitrate over European polluted regions, such as northern polish region, and a clear reduction of AOD_nitrate over most of European continent (see Fig. 5a). For US case, we see outflows bring continental pollutants to marine and a clear 're-distribution effect' leading to reduction of AOD_nitrate over oceanic regions, such as the Gulf of Mexico; but, there is much weaker 're-distribution effect' over the N. American continent (see Fig. 8a). This difference between the European and US cases indicates that 're-distribution effect' happens wherever natural sea-salt and anthropogenic nitrate mixed, no matter over continental or oceanic regions. Lots of big cities are along coast globally with high NOx (precursor of nitrate) emissions, offshore and onshore wind patterns can both induce the 're-distribution effect' and reduce AOD_nitrate.*

*The similarity. Although we see different spatial patterns of the 're-distribution effect' in the European and US cases, the relationships between AOD_nitrate reduction and RNS (molar ratio*

*between nitrate and sodium) are quite consistent in these two cases. As suggested by the Reviewer#2, we show this relationship as a first-order approximation, given in the newly added Fig. 9b. Later on, we adopt this generalized relationship combined with EMAC global model results to estimate the potential changes of $AOD_{nitrate}$ associated with 're-distribution effect' on a global scale.*

[Figure]

***Newly added Fig. 9b.*** *The median possibility of the percentage change of $AOD_{nitrate}$ as a function of RNS.*

3) On p.4 l.116 the authors state that they found in previous studies that – using the Gong emission parameterization – modeled sea-salt concentrations overestimated measurements by a factor of 10. However, the sea salt emissions estimated by the Gong parameterization are not necessarily 10x as high as the real world sea salt emissions are. Processes that lead to lower atmospheric sea salt concentrations might be underestimated in the used model – e.g. underestimated deposition. Or, the particle size distribution of the Gong sea salt emissions might not be appropriate for the model setup. Therefore, please add "… using WRF-Chem …" or something similar to the sentence – e.g. in the end of the sentence.

*We have modified the context as suggested.*

4) In the first paragraph on page S9 in the supplement, the authors discuss that nitrate concentrations are overestimated by several models in Europe. Vivanco et al. (2017, DOI: https://doi.org/10.1016/j.atmosenv.2016.11.042) evaluated several air quality models and found that most considered models underestimate the nitrogen deposition in Europe (The

EMEP model performed best). WRF-Chem was not partof the latter evaluation. However, too high atmospheric nitrate concentrations may point to issues in the deposition parameterizations. This wouold also explain why the Gong parameterization apparently lead to considerably overestimated atmospheric sea salt concentrations. Could the authors comment on that?

*This is a good point, thanks for the suggestion. We have added a comment in the last paragraph of section 3.4, as shown below.*

*"The uncertainty in deposition parameterizations could be one reason of this overestimation, as suggested by the overestimation of sea-salt particles as well (Chen et al., 2016a) and the underestimation of nitrogen deposition over Europe in many models (Vivanco et al., 2017)."*

5) Evaluating the mass concentration of sea salt aerosol is reasonable to evaluate the coarse sea salt emissions. The contribution of fine sea salt to the total sea salt mass concentration is very low. Hence, the strong overestimation of sea salt mass concentrations documented in previos studies does not necessarily indicate that the fine particulate sea salt was overestimated as well. The black bars in Fig. S3 rather indicates that fine sea salt might be underestimated in some episodes. Please briefly describe this uncertainty in the Materials & Methods or in the Discussion section.

*Good point. We have added discussion of this uncertainty in the Materials & Methods, section 2.2, as shown below.*

*"The sea-salt emissions computed with the modified Gong scheme (Gong, 2003) were reduced to 10% in the 'Case_SeasaltOn', because a previous study (Chen et al., 2016) has shown that the original Gong scheme overestimates the sea-salt mass concentrations by a factor of ~10 over the coastal regions of Europe using WRF-Chem model. We note that although the mass of coarse sea-salt particles is certainly overestimated, it might not necessarily indicate overestimation in fine sea-salt particles related to their minor contribution to the total mass."*

6) p.6 l.165-168 "In addition, a one-year simulation with global model (EMAC) was carried out for analysis of the potential impact of 're-distribution effect' on a global scale, although the fully dynamic mass transfer between particle sizes is not considered in EMAC (four size modes

rather than eight size bins as applied in the WRF-Chem model)." Why is EMAC neverthelss applicable for the used purpose?

*Here, we did not directly use EMAC model to estimate the impact of 're-distribution effect'. We generalized a relationship between the impact of 're-distribution effect' and RNS (molar ratio between nitrate and sodium) from the European and US cases, and then used this relationship combined with the RNS values from EMAC model to investigate the potential impact of 're-distribution effect' on a global scale. Please also see the 'The similarity' part of the response in point-2. We have modified the context to make it clearer, as shown below.*

*"In addition, a one-year simulation with global model (EMAC) was carried out for analysis of the potential impact of 're-distribution effect' on a global scale. Although the fully dynamic mass transfer between particle sizes is not considered in EMAC (four size modes rather than eight size bins as applied in the WRF-Chem model), we adopt a parameterization derived from WRF-Chem simulations to estimate the potential impact (details given in section 3.5)."*

7) "Sea-salt is emitted into the marine planetary boundary layer (PBL) as coarse particles [. . . ]" (p.7 l.183). Aren't it coarse and fine particles? Depending on the considered moment of the particle size distribution the fine or the coarse mode dominates: the number distribution is dominated by the fine mode; the volume/mass distribution is dominated by the coarse mode. Please clarify this in the text.

*Yes, the reviewer is right that it has to be precise. We have clarified this, as shown below.*

*"Sea-salt is emitted into the marine planetary boundary layer (PBL) with mass concentration dominated by coarse particles…"*

8) The authors mention the importance of vertical transport of sea salt aerosol into the continental free troposphere followed by the horizontal transport of it over land and refer to their previous studies (p.7 l.186-190). This transport mechanism seems to be very important. However, vertical transport/convection is not necessarily well represented by all meteorological models – depending on grid resolution and parameterization. This study's

WRF-Chem model simulations probably reproduce this vertical transport of sea salt? Do the other used models reproduce it as well?

*The vertical transport of sea-salt particles to free troposphere can increase their lifetime and facilitate their inland transport, therefore, it could be important for the 're-distribution effect' over the deeper inland regions. This vertical transport and the associated land inward transport are reproduced well in our high-resolution simulation using WRF-Chem, as shown in our previous study (Chen et al., 2016). However, as the reviewer concerned, this vertical transport may not be well reproduced in other models with lower resolutions. But, this does not necessarily mean that sea-salt cannot transport further inland in other models. The long-range transport of sea-salt to inland regions also depends on the deposition scheme (which might be underestimated as pointed out by the reviewer in the point-4), the vertical mixing rate, the parameterizations for marine boundary layer and continental boundary layer and etc. These can be different model-by-model and case-by-case, and the discussion of these differences go beyond the scope of this study.*

*But, I think the vertical transport might not be such important for the coastal regions, where lots of big cities with high NOx emissions are located, especially for China and N. America with many large and the most developed cities along the coast. This is because no matter offshore or onshore wind patterns both mix sea-salt with anthropogenic nitrate and induce the 're-distribution effect', as discussed in the point-2.*

9) In chapter 3.3 (p.10 l.281-290), the authors mention the aim to 'generalize' the results. To 'generalize' something has a quite broad meaning and can be interpreted differently. The benefit of calculating the RNS and of including Fig. 5 did not become clear for me in chapter 3.3 – but, later in chapter 3.4 it became obvious. Please clarify the motivation in chapter 3.3.

*We have modified the corresponding sentence and clarify the motivation in section 3.3, as shown below.*

*"To compare the relative importance of the sea-salt-induced 're-distribution effect' and 'mass-enhancement effect' on anthropogenic nitrate cooling…"*

10) Classification of RNS: Currently, we have the situations "RNS < 1", "1 < RNS < 30" and "30 < RNS". I know that these are rough classifications. Nevertheless, please also cover "RNS == 1" and "RNS == 30" to be mathematically correct.

*Thanks for the carefulness. We have corrected it as suggested.*

**Comments on figures:**

1) Figure 1: I like the intention of Fig. 1. All relevant aspects are shown. However, on the first view, the reader might not recognize the particle size distribution plot as such. At least I did not recognize it first. I am not sure how to improve the Figure. Possibly, a y-axis on the left of the plot might help. Their are no issues when the figure is printed in grey-scales (see my comments to the other figures below).

*We have added a y-axis in Fig. 1. It does make the particle size distribution plot clearer. Thanks.*

2) Figure 2, 6, 8 and 9: Please consider using a color scale, which is recognizable in black-and-white and readable by color blind people (not "jet" or "rainbow"). Examples for such a color scale are "viridis" and "magma".

*We have changed the color scales of Fig. 3, Fig. 7, Fig. 9 and Fig. 10 (Fig. 2, Fig. 6, Fig. 8 and Fig. 9 in the original version) to make them recognizable in grey-scales.*

3) Figure 4 and 7: Printed in grey scales it is hard to distinguish whether values are negative or positive. However, alternative color scales would make the full-color plots difficult to interpret. Therefore, it might be reasonable to keep the blue-white-red colorscale.

*We did not find a proper color-scales fit in the grey-scales perfectly, and also clearly show the trends. Agree with the reviewer, it is better to keep it as it is.*

4) Figures 3: Please add information to the plot that nitrate is plotted. It is written in the caption but it would be nice to also have this information in the plots. The colors in this figure are fine.

*Done. 'Nitrate' has been added in the y-axis.*

5) Figures 5: The colors in this figure are fine. Figure S6: Maybe exchange panels (a) and (b). For me it would have made the figure easier to understand. I would consider adding Figure S6 to the main manuscript.

*Thanks. As suggested, we have moved the Fig. S6a (original) to Fig. 7c (revised), and Fig. S6b (original) to Fig. 7b (revised).*

**Minor Comments:**

p.4 l.117: consider to replace "for both with and without sea-salt presence respectively" by "for both sea salt emission cases"

p.5 l.123: remove "and" in ". . . assumption and taking . . . "

p.5 l.136: "and part of North Africa" → "and the northern part of North Africa"

p.5 l.142 "1/8_ × 1/16_" → space between "×" and "1/16"

p.5 l.147 "In the 'Case SeasaltOn' (with sea-salt emission) of European simulation" → "In the European 'Case SeasaltOn' simulation (with sea-salt emission)", suggestion

p.5 l.148 "over coastal region" → "over coastal regions", added plural-s

p.6 l.149/150 "with a factor (and correlation coefficient) of 0.85 (0.67), 1.16 (0.80) and 0.83 (0.87) respectively for Bilthoven, Kollumerwaard and Vredepeel (Fig. S2)" → "with a factor (and correlation coefficient) of 0.85 (0.67), 1.16 (0.80) and 0.83 (0.87) for Bilthoven, Kollumerwaard and Vredepeel, respectively (Fig. S2)", moved 'respectively'

p.6 l.154 "over coastal, German low lands (Melpitz) and northern Poland regions" → "over coastal, German low land (Melpitz) and northern Polish regions", remove 's' from lands because we have "coastal regions", "German low land regions" and northern Polish regions"

p.6 l.162 "concentration of nitrate was" → "concentrations of nitrate were"

p.6 l.164 "covers US, the Gulf of Mexico and part of Pacific and Atlantic oceans" → "covers the US, the Gulf of Mexico and parts of Pacific and Atlantic oceans", 'the' in front of 'US' and 's' added to 'part'

p.6 l.171 "1.1 by 1.1 degrees" → "1.1º × 1.1 º", formatting consistent with p.5 l.142

p.6 l.161 "boarder" → "broader"

p.6 l.177 "approach Central Europe, and the interaction" → "approach Central Europe.

The interaction"

p.7 l.179 "a typical sea-salt transport event"; maybe 'sea salt transport event' in italics

p.7 l.182 "originated" → "originating"

p.7 l.187 "long-range transport, see the figure 11 of Chen et al. (2016a)" → "long-range transport (see Fig. 11 of Chen et al. (2016a))"

p.8 l.212: "the northern Poland region" → "a region in northern Poland"

p.8 l.220: ",thus " → ", thus, "

p.8 l.227: "Over the northern Poland region" → "Over the region in northern Poland"

p.8 l.231-234: "being slightly . . . in the offline calculation" – Please consider writing this sentence outside of the brackets. It improves the readability.

p.9 l.241-242: "with increasing AOD with [NO3-]" → "of increasing AOD with increasing [NO3-]"; I am not sure about it but for me it sounds better.

p.9 l.242: "would result by neglecting the 're-distribution effect'" → "would result when t he 're-distribution effect' was neglected"

p.9 l.242-244: "For example, instead of a decrease by 29

p.9 l.245: remove "also"

p.9 l.245: ". . . increases and there is . . . " → ". . . increases. There is . . . "

p.9 l.248: "and thus shorter" → "and, thus, shorter"

p.9 l.251: "and hence further moderate" → "further moderating"

p.9 l.254: ", therefore fasten" → "fastening"

p.9 l.255: "oxygenated nitrogen" → maybe "oxidized nitrogen"?

p.9 l.257: "which result" → "which resulted"

p.9 l.258-263: This has been mentioned previously. Although, this is important motivation for performing another simulation without deposition, I would like to suggest shortening these sentences considerably.

p.10 l.265-266: "as the simulations with aerosol dry deposition" → "as in the simulations with aerosol dry deposition turned on"; corresponds better with l.263-264 stating ". . . with aerosol dry deposition turned off"

p.10 l.289: "due to" → "because"; 'due to' does not allow a verb ('are')

p.11 l.319-320: "Myhre et al. (2006), where a similar . . . aerosol was evaluated but the simplification . . . " → "Myhre et al. (2006). They evaluated a similar . . . aerosol. But, their simplification . . . "; split into two or three individual sentences;

p.12 l.332: "and thus reduces" → "and, thus, reduces"

p.12 l.337: "presented" → "present"

p.12 l.342-343: "that sea-salt transport (May et al., 2018) and impact AODnitrate further inland over North America." → "that sea-salt is transported further inland over North America und impacts AODnitrate there (May et al., 2018)."

p.13 l.352-356: Please refer to Fig. 9 (maybe I overlooked the reference).

p.13 l.360: "and hence lower" → "and, hence, lower"; I would consider splitting the sentence into two sentences at this 'and'.

p.13 l.369: "if only consider" → "if we only consider" or passive form

p.13 l.374: ". . . when RNS is lower than 1" → Formulation is ambiguous in this context: the dominance could take place when RNS < 1 OR the mass-enhancement effect could be associated with RNS < 1. It know (and probably most readers) know that the latter is meant. However, I would suggest to reformulate this part. The same for the next sentence (". . . when 1 < RNS < 30").

*Many thanks for the careful reading, helping correct the typos and improving the language. We have improved the manuscript as suggested. Please find details in the change-tracked revised manuscript.*

**References:**

Birmili, W., Wiedensohler, A., Heintzenberg, J., and Lehmann, K.: Atmospheric particle number size distribution in central Europe: Statistical relations to air masses and meteorology, Journal of Geophysical Research, 106, 32005-32018, DOI: 10.1029/2000JD000220, 2001.

Chen, Y., Cheng, Y., Ma, N., Wolke, R., Nordmann, S., Schüttauf, S., Ran, L., Wehner, B., Birmili, W., van der Gon, H. A. C. D., Mu, Q., Barthel, S., Spindler, G., Stieger, B., Müller, K., Zheng, G. J., Pöschl, U., Su, H., and Wiedensohler, A.: Sea salt emission, transport and influence on size-segregated nitrate simulation: a case study in northwestern Europe by WRF-Chem, Atmos. Chem. Phys., 16, 12081-12097, 10.5194/acp-16-12081-2016, 2016.

Gong, S. L.: A parameterization of sea-salt aerosol source function for sub- and super-micron particles, Global Biogeochemical Cycles, 17, 10.1029/2003GB002079, 2003.

Vivanco, M. G., Bessagnet, B., Cuvelier, C., Theobald, M. R., Tsyro, S., Pirovano, G., Aulinger, A., Bieser, J., Calori, G., Ciarelli, G., Manders, A., Mircea, M., Aksoyoglu, S., Briganti, G., Cappelletti, A., Colette, A., Couvidat, F., D'Isidoro, M., Kranenburg, R., Meleux, F., Menut, L., Pay, M. T., Rouïl, L., Silibello, C., Thunis, P., and Ung, A.: Joint analysis of deposition fluxes and atmospheric concentrations of inorganic nitrogen and sulphur compounds predicted by six chemistry transport models in the frame of the EURODELTAIII project, Atmospheric Environment, 151, 152-175, https://doi.org/10.1016/j.atmosenv.2016.11.042, 2017.

---

## Author Comment (AC2) · 16 Nov 2019

**Response to comments of referee#2**

**General comments:**

This is a well-written paper that highlights an interesting topic that I had not previously thought about, but makes a lot of sense conceptually and could have important implications for anthropogenic aerosol radiative forcing on a global scale. I have one comment which questions some of the methods used in the experiment, which I am not expecting the authors to address in this paper but could be the subject of further discussion or future work. Beyond that, I just have a few minor comments I think the authors should

*Many thanks to the reviewer for the insightful comments and suggestions. We have improved the manuscript accordingly. Please find a point-by-point response below. Please refer the order of figures to the revised version.*

**Main comments:**

My main comment is that the main purpose of this paper is to investigate the sensitivity of the nitrate aerosol radiative forcing to the process that redistributes it from fine modes when associated with ammonium to course modes when associated with sodium. The authors have tested this by turning off sea salt emissions in the sensitivity simulation and seeing how this changes nitrate aerosol size distribution, which is perfectly valid but does limit what they can investigate in some ways because removing sea salt emissions will have many other changes to total aerosol AOD, CCN and aerosol-radiation/aerosol-cloud interactions that go beyond the redistribution effect they are investigating. I would have thought a better experiment would have been to turn off the association between NO3 and Na in MOSAIC in the sensitivity simulation, and have sea-salt emissions in both, thereby having a "clean" experiment where the only change is to the exact process you are investigating. A key benefit of this approach is that you would then be able to test the impact of this process on both total aerosol AOD/DRF (not just the NO component), and how it effects the aerosol-cloud interactions (ACI).

On both a local and global scale, my gut says that the influence of the redistribution effect on (ACI), particularly the Twomey or first- indirect effect, would be greater that the DRF. I would

expect shifting nitrate from subnitrate modes to coarse modes would slightly increase the number of low-supersaturation CCN, whilst massively decreasing the number of high-supersaturation CCN. This could in turn have a large impact on the activated cloud droplet number, cloud optical depth and therefore climate. Unfortunately, the current methodology cannot address this as any impact from the redistribution effect would be likely overwhelmed by the lack of sea salt aerosol in the sensitivity simulation. This is a shame, as without investigating the aerosol-cloud interactions we are only seeing part of the impact of this process on the climate.

As I said above, I'm not expecting the authors to do reruns with this change as it is a substantial amount of work and, to my knowledge, their current paper is already the first I know of to look into this problem in detail and already contributes significantly to the field. But I would be interested in hearing their response and any follow up studies that look into this further – this discussion could be included in the revised paper.

*Thanks for the positive comments on the scientific meaning of this study and insightful suggestions. We agree with the reviewer that turning off the association between NO3 and Na in MOSAIC can be an optional to produce 'clean' sensitivity experiments for investigating the influence of re-distribution on radiative forcing (RF). I think the method we used in this study also generates the 'clean' experiments for the influence of re-distribution on direct RF$_{nitrate}$, although the review's suggestion can be a better way for investigating ACI and indirect RF$_{nitrate}$.*

*What we did in this study is not simply turn off the sea-salt emissions and compare the AOD between simulations with and without sea-salt. As correctly pointed out by the reviewer, this would include lots of noise from the changes of sea-salt. Alternatively, we performed four sensitivity simulations (cases 01-04) to derive the 'clean' influence of nitrate re-distribution on AOD$_{nitrate}$, which was provided in the Method (line: 127-130 in the change-tracked version). Nitrate AOD and DRF (direct RF) with 're-distribution effect' is calculated as: differences between with (case 01) and without (case 02) the anthropogenic emitted gas phase precursor NOx in the simulations with sea-salt emissions turned on. Nitrate AOD and DRF without 're-distribution effect' is calculated as: differences between with (case 03) and without (case 04) the anthropogenic emitted gas phase precursor NOx in the simulations with sea-salt emissions turned off. This calculation method is in line with IPCC and previous studies (IPCC, 2013;Xu and Penner, 2012). For the impacts of the 're-distribution effect', we calculate the differences between nitrate AOD/DRF with 're-distribution effect' (case01-case02) and nitrate AOD/DRF*

*without 're-distribution effect' (case03-case04). In this approach, the influences of changes in sea-salt on AOD and DRF were ruled out.*

*But, we agree with the reviewer that our method can not rule out the noise of sea-salt influence on ACI and IRF (indirect RF). Therefore, in this study, we only analyzed the results which are marked as cloud-free (i.e. cloud optical depth equals to zero, see Method section in line 130-131 of the change-tracked version) in cases 01-04. In this study, we only focus on the influence of 're-distribution effect' on the direct radiative forcing of nitrate. However, we agree with the reviewer that the influences on ACI and IRF is another important impact of the 're-distribution effect' and might be even larger. This question needs to be investigated in further studies, possibly with the method suggested by the reviewer. We have added the discussion about this point at the end of the section 4, as shown below.*

*"This study demonstrates the suppression of AOD and DRF of particulate nitrate by the 're-distribution effect'. In addition, the 're-distribution effect' may also reduce the number of cloud condensation nuclei (CCN) by lowering the nitrate concentration in fine particles which are the main contributors to CCN number. The hygroscopicity of coarse sea-salt particles could also be reduced by associating with nitrate, which might suppress cloud droplet activation (Xu and Penner, 2012). Further studies are needed to investigate the influences of 're-distribution effect' on aerosol-cloud interaction and indirect radiative forcing."*

**Minor comments**

1) Ln 40-41: Under what kinds of emission scenarios are we expecting the aerosol radiative cooling to increase by this much?

*According to the provided references, there are a series of different emission scenarios used in the projection, such as IPCC SRES (Special Report on Emission Scenarios) A2 emission Scenario in Adams et al. (2001), RCP8.5 (Representative Concentration Pathways) in Bellouin et al. (2011) and RCP CMIP5 (Climate Model Intercomparison Project) emission scenarios in Hauglustaine et al. (2014).*

2) Ln 76: Please explain acronym for HOPE-Melpitz campaign

*We have modified the sentence as shown below.*

*"The HOPE campaign (HD(CP)2 Observational Prototype Experiment, (Macke et al., 2017) at Melpitz, Germany"*

3) Ln 82: "which represent continental period and marine period" Is this based from back trajectories? Please explain better.

*Yes, it is based from back-trajectories. We have added this information in the context and the back-trajectories in the supplementary information Fig. S1. As shown below.*

*"which represent the continental period and marine period, respectively (see back-trajectories in Fig. S1)."*

[Figure]

*Fig. S1. Three days back-trajectories for Melpitz, Germany. (a) 13 September 2013, representing continental period; (b) 18 September 2013, representing marine period. The back-trajectories are calculated by Hysplit (https://www.ready.noaa.gov/HYSPLIT.php).*

4) Section 2.2 Model Description: this section is missing some important information. Firstly, I couldn't find anywhere a description of how the MOSAIC mechanism describes the chemical uptake of HNO3 onto aerosol via NH4+ or Na+ channels. A short (onetwo sentences) here describing that process and how it is parameterised in MOSAIC would be welcome.

*We have added a brief summary in section 2.2 to introduce that how the MOSAIC describes the chemical uptake of $HNO_3$ onto aerosol via $NH_4^+$ and $Na^+$ channels. As shown below.*

*"In MOSAIC, NaCl reacts irreversibly with nitric acid with its equilibrium surface vapor pressure of zero; and a gas-particle partitioning approach ASTEM (Adaptive Step Time-Split Euler Method) is coupled with a thermodynamic module (MESA-MTEM, Multicomponent Equilibrium Solver for Aerosols – Multicomponent Taylor Expansion Method) to dynamically calculate the equilibrium vapor pressure and condensation rate of semi-volatile ammonium nitrate, details are given in the section 4 of (Zaveri et al., 2008)."*

5) Please can you add what meteorology data you are using to drive the model and if you are nudging, or if it freerunning (with or without feedbacks). Some of this information is in the supplement.

*We have added this information in the supplement Section S2, as shown below.*

*"The WRF-Chem model was driven by NCEP reanalysis data ($1^o \times 1^o$ resolution and provided every 6 hours), including the Final Analysis Operational Global Analysis (http://rda.ucar.edu/datasets/ds083.2/) and the sea surface temperature datasets (http://polar.ncep.noaa.gov/sst/). The first two days were spin-up runs for simulations in this study. The nudging is carried out in every 6 hours for meteorological conditions, including wind, temperature and moisture."*

6) Finally, are you running with N2O5 heterogeneous chemistry? This is relevant for a point later on in the paper, plus in WRF-Chem the Bertram and Thornton (2009) parameterization does have a channel for uptake via Cl-, so provides another mechanism for nitrate to be taken up to course modes that is modulated by the presence of sea salt (Archer-Nicholls et al., 2014).

Bertram, T. H., & Thornton, J. A.,Toward a general parameterization of N2O5 reactivity on aqueous particles: the competing effects of particle liquid water, nitrate and chloride. Atmospheric Chemistry and Physics, 9(4), 15181–15214. https://doi.org/doi:10.5194/acp-9-8351-2009, 2009.

Archer-Nicholls, S., Lowe, D., Utembe, S., Allan, J., Zaveri, R. A., Fast, J. D., et al., Gaseous chemistry and aerosol mechanism developments for version 3.5.1 of the online regional model, WRF-Chem. Geoscientific Model Development, 7, 2557– 2579. https://doi.org/10.5194/gmd-7-2557-2014, 2014

*In this work, we did not consider $N_2O_5$ hydrolysis with NaCl, we focus on the uptake of $HNO_3$ because this process makes fine mode nitrate particles shift to the coarse mode, i.e., the 're-distribution effect'. The $N_2O_5$ reaction is important during night. Due to the topic of radiative forcing in this study, we are mainly focusing on the daytime period. But, we agree with the reviewer that $N_2O_5$ hydrolysis with NaCl is an important pathway of particulate nitrate in coarse mode, the considering of this process may make the 're-distribution effect' stronger. We have added this comment in section 2.2, as shown below.*

*"We note that heterogeneous hydrolysis of $N_2O_5$ with NaCl is an important chemical pathway of particulate nitrate in coarse mode during nighttime (Bertram and Thornton, 2009; Archer-Nicholls et al., 2014). This process may enhance the 're-distribution effect', however it is not considered in this study."*

7) Ln 110: Is the version of MOSAIC being used with or without aqueous chemistry?

*The version of MOSAIC is being used with some aqueous chemistry, as described in Zaveri et al. (2008). However, $N_2O_5$ hydrolysis and the oxidation of $SO_2$ in aqueous aerosols are not included.*

8) Ln 137: Please include figure of outer and inner domains, here or in supplement.

*We have added a map of the outer and inner domains in the supplement Fig. S2.*

[Figure]

**Fig. S2.** *Domain setting of WRF-Chem European case.*

9) Ln 249: In terms of lifetime of nitrate radical (i.e. gas phase NO3), I would have thought this has more to do with the uptake of N2O5 to aerosol enhanced by Cl- rather than the HNO3 process you discuss in this paper. I'm not sure if it is quite relevant with the discussion here, which is more about the lifetime of aerosol NO3- when it is in different aerosol size modes. Please clarify.

*Thanks for the comment. We agree with the reviewer that the discussion here might be more relevant with the uptake of N2O5 to NaCl rather than the lifetime of particulate nitrate in different aerosol size modes. We therefore removed the discussion here to make the point clearer.*

10) Ln 336-347: Would it be possible to show a map of RNS over Europe and North America from the WRF-Chem model for comparison? You've only done this for the global EMAC

model, which by the sounds of things has lower values for RNS over most of the US compared to WRF-Chem (essentially nowhere in North America has RNS > 30 in the EMAC model, whereas this regime dominate continental North America in the WRf-Chem simulations). The figures currently given make it hard to follow the logic in this paragraph. In general, I think this section can be improved by outlining the conclusions that can be drawn from the evidence given, rather than speculating on what we might expect the effects to be if sea salt is transported further inland.

*We have provided the map of RNS (molar ratio between nitrate and sodium) over Europe and North America from the WRF-Chem model as below (Fig. R1). The reviewer is right that RNS values from WRF-Chem cases are generally higher than the values from EMAC global model. However, these results are not directly comparable, because the EMAC global model provides the information of annual average but WRF-Chem provides results of a short period. In this paper, we would like to generalize a relationship between RNS and changes of $AOD_{nitrate}$ due to the 're-distribution effect' from WRF-Chem results; and then estimate the potential impact of the 're-distribution effect' on a global scale, by adopting this relationship combined with RNS values from EMAC global model (as correctly pointed out by the reviewer in the next comment). Therefore, we prefer not to include Fig. R1 in the manuscript, because it is not directly comparable with the results of the EMAC model and does not clearly show the relationship. Alternatively, we have added a Fig. 9b to clearly show the relationship between RNS and changes of $AOD_{nitrate}$ as a first-order estimate, as suggested by the reviewer in the next comment. This makes the discussion clearer.*

*Thanks for the suggestion about how to improve the logic of this paragraph. We have re-written this paragraph as suggested, and moved the discussion of differences between Europe and North America and further transport of sea-salt to the next paragraph, as shown below.*

*"The statistical analysis of the 're-distribution effect' over North America (Fig. 9a) shows a similar pattern as over Europe (Fig. 7a), and a first-order approximation ($R^2 > 90\%$) is derived from the European and North American results of WRF-Chem model to parameterize the relationship between RNS and the changes of $AOD_{nitrate}$ associated with the 're-distribution effect' (Fig. 9b). In general, the impact of the 're-distribution effect' on $AOD_{nitrate}$ decreases as RNS increases. Only considering the 'mass-enhancement effect' but ignoring the 're-*

*distribution effect' may lead to an overestimation of $AOD_{nitrate}$ by about 20% when RNS < 1, by about 10-20% when $1 \leq RNS \leq 30$ and by less than ~10% when RNS > 30.*

*On a global scale, the potential influence of the 're-distribution effect' on $AOD_{nitrate}$ is estimated by using the above first-order approximation combined with a one-year RNS simulation with the EMAC (Klingmüller et al., 2014;Pringle et al., 2010) chemistry-climate model (Fig. 10). The global distributions of percentage changes of $AOD_{nitrate}$ and surface nitrate concentration are given in Fig. S8. In line with the WRF-Chem results, a significant 're-distribution effect' is expected over North America and Europe, especially over the coastal regions with high nitrate loading and RNS values around 1 (Fig. 10b). As shown in Fig. 10c, the impact over Europe is stronger than over North America. The oceanic influence dominates over western Europe whereas over North America the predominantly more continental air masses (Kottek et al., 2006) may be a possible reason of this. It leads to less interactions of sea-salt with anthropogenic nitrate in North America, and more significant reduction of $AOD_{nitrate}$ over Europe. This result is consistent with a previous study (Myhre et al., 2006). Nevertheless, it is still possible that sea-salt is transported (May et al., 2018) further inland over North America and impacts $AOD_{nitrate}$ there. The coastal and offshore regions of Asia with $1 \leq RNS \leq 30$ may experience strong reductions of $AOD_{nitrate}$, where the 're-distribution effect' is expected to overwhelm the 'mass-enhancement effect', such as coastal and outflow regions of China. The influence of sea-salt aerosol becomes negligible over inland Asia where marine air mass influence is small (RNS > 30). For the open sea regions with RNS < 0.01 (white background), nitrate climate effect is not important, due to very limited nitrate loading contributing to $AOD_{nitrate}$ (Fig. 10a)."*

[Figure]

[Figure]

***Figure R1.*** *Map of RNS (molar ratio between nitrate and sodium) in the European case (a) and the North American case (b) from the WRF-Chem simulation. Note that the color-scales have been changed to grey-scale friendly style, as suggested by the Reviewer#1.*

[Figure]

[Figure]

**Fig. 9. Intensity of 're-distribution effect' as a function of molar ratio between surface fine nitrate and total sodium (RNS). (a)** *The intensity of 're-distribution effect' over North American domain, similar as Fig. 7a, calculated as the difference in percentage between $AOD_{nitrate}$ and $AOD^*_{nitrate}$ in the 'Case_SeasaltOn'. $AOD^*_{nitrate}$ indicates the $AOD_{nitrate}$ calculated by re-allocating nitrate mass into different size bins according to the normalized nitrate particle mass size distribution simulated in 'Case_SeasaltOff' (i.e., without 're-distribution effect'). The black dots indicate the mean values; the upper and lower error bars indicate the 75% and 25% percentile, respectively. The colour indicates the frequency distribution (i.e., how many counts) of the hourly model results over entire North America domain during 10-17 January 2015.* **(b)** *The median possibility of the percentage change of $AOD_{nitrate}$ as a function (first-order approximation) of RNS.*

11) Ln 348-350: Would it be possible to plot a first-order estimate of what the change to nitrate AOD would be from this effect, using a function based on the relation between RNS and [NO3-] derived from the WRF-Chem model simulations?

*This is a good point. We have plotted the first-order approximation in the newly added Fig. 9b, provided the estimated change of nitrate AOD associated with the 're-distribution effect' in the Fig. 10c, and added the corresponding discussions in the last two paragraphs of section 3.5. Please find details in the response to the last comment. This does make the discussion clearer. Thanks for your suggestion.*

12) Ln 364: "(_90%)" – what is this referring to exactly?

*We have modified the description to make this clearer, as shown below.*

*"up to 90% of the days in a year"*

13) Ln 390-394: This last section is confusing what you can derive from the results of this study (modulation to nitrate DRF from sea-salt aerosol) from that speculated from other related chemical interactions (sulfur, dust etc.). Please rewrite to clearly separate the conclusions drawn from your results from speculations/highlight topics for future research.

*We have re-written the context in the last section to clearly separate the conclusions drawn from this study from highlight topics for future research, as shown below.*

*"This study highlights the impact of the 're-distribution effect' on moderating nitrate cooling and altering the nitrogen deposition efficiency by interacting with natural sea-salt aerosols (Fig. 1) … All these previous studies imply the possibility that natural particles (sea-salt aerosol and very likely dust as well) moderate the DRF of anthropogenic aerosols and alter the nitrogen and sulfur deposition efficiency. We highlight the importance of further study of the inter-actions between natural and anthropogenic aerosols."*

**Figures:**

1) Please promote Fig S1 into the main paper, I think it really clearly shows how the redistribution effect changes the aerosol chemical size distribution.

*Modified as suggested.*

2) Fig 9. Please can you include a subplot with this showing [NO3-]? You need this with RNS to get an understanding of where the redistribution impact is likely to be strongest.

*We have added the map of [NO$_3^-$] in the Fig. S8b, as shown below. Please also see detailed discussion in the Minor Comment-10.*

[Figure]

*Fig. S8b. Global distribution of percentage changes of surface nitrate concentration. The results are from EMAC model.*

**References:**

[revised manuscript text omitted]